# Performance of Low-Dimensional Solid Room-Temperature Photodetectors—Critical View

**DOI:** 10.3390/ma17184522

**Published:** 2024-09-14

**Authors:** Antoni Rogalski, Weida Hu, Fang Wang, Piotr Martyniuk

**Affiliations:** 1Institute of Applied Physics, Military University of Technology, 2 Kaliskiego Str., 00-908 Warsaw, Poland; antoni.rogalski@wat.edu.pl; 2State Key Laboratory of Infrared Physics, Shanghai Institute of Technical Physics, Chinese Academy of Sciences, 500 Yu Tian Road, Shanghai 200083, China; wdhu@mail.sitp.ac.cn (W.H.); fwang@mail.sitp.ac.cn (F.W.)

**Keywords:** BLIP, SFL, photogating effect, ultraviolet photodetectors, 2D photodetectors, HOT photodetectors, topological insulator photodetectors, colloidal quantum dot photodetectors, perovskite photodetectors

## Abstract

In the last twenty years, nanofabrication progress has allowed for the emergence of a new photodetector family, generally called low-dimensional solids (LDSs), among which the most important are two-dimensional (2D) materials, perovskites, and nanowires/quantum dots. They operate in a wide wavelength range from ultraviolet to far-infrared. Current research indicates remarkable advances in increasing the performance of this new generation of photodetectors. The published performance at room temperature is even better than reported for typical photodetectors. Several articles demonstrate detectivity outperforming physical boundaries driven by background radiation and signal fluctuations. This study attempts to explain these peculiarities. In order to achieve this goal, we first clarify the fundamental differences in the photoelectric effects of the new generation of photodetectors compared to the standard designs dominating the commercial market. Photodetectors made of 2D transition metal dichalcogenides (TMDs), quantum dots, topological insulators, and perovskites are mainly considered. Their performance is compared with the fundamental limits estimated by the signal fluctuation limit (in the ultraviolet region) and the background radiation limit (in the infrared region). In the latter case, Law 19 dedicated to HgCdTe photodiodes is used as a standard reference benchmark. The causes for the performance overestimate of the different types of LDS detectors are also explained. Finally, an attempt is made to determine their place in the global market in the long term.

## 1. Introduction

Lately, a novel family of compounds, usually referred to as low-dimensional solids (LDSs), has emerged to contend with those typically available on the market—commercial photodetectors; among them, two-dimensional (2D) materials and nanowires (NWs)/quantum dots (QDs) may be listed [1]. Since 2D materials exhibit novel and unusual properties, they have been found to be promising for high-performance devices:Energy states are easily tuned by external fields (gate-induced electrostatic fields, photogating localized fields, ferroelectric fields) for materials with a typical thickness <10 nm.The energy bandgap ranges from 0 to 6 eV (for graphene and hexagonal boron nitride, h-BN, respectively), making it possible to fabricate photodetectors operating over a broad spectral range from ultraviolet (UV) to far-infrared (FIR) and terahertz (THz).The energy bandgap is closely related to thickness—increasing the thickness lowers the bandgap.There is a direct bandgap for MoSe_2_, MoS_2_, WSe_2_, and WS_2_ [transition metal dichalcogenides (TMDs)] and an indirect bandgap for bulk materials.

Compared to typical semiconductors, the fabrication of 2D material-based devices (random stacking) has been found to have a significant advantage in reducing the lattice mismatch between potential substrates and active layers. The stack of atomic planes is coupled by van der Waals (vdW) interactions without dangling bonds, allowing for the deposition of vertical structures and the implementation of 2D compounds into Si technology. Moreover, vdW forces enable the large-scale deposition of 2D compounds independent of potential substrates. Finally, the compatibility of large-scale 2D materials with Si technology allows for great prospects for the fabrication of optoelectronic devices.

Unfortunately, at the current technology level, the following problems hinder the development of 2D compound-based detectors:Susceptibility to absorbates (defects and unintentional doping).Performance driven by layer stacking (spacing, twist) and environment (pressure, strain).Poor linear dynamic range (LDR)—lack of applicability to intense light caused by atomically thin nature (despite the large absorption coefficient).

In spite of the listed problems, a new group of materials (2D and QDs/NWs) has emerged that may contend with typical photodetectors available on the market. In addition, numerous papers have reported on this subject, several being exceedingly enthusiastic. To date, some of the published papers have shown their results being misread due to the problem of accurate device characterization [2,3,4,5].

This article focuses on ~300 K detectors, as significant efforts are currently being made to raise the operating temperature and meet the size, weight, and power consumption (SWaP) conditions and reduce costs, especially for the infrared (IR) spectral range. The fundamental purpose of these attempts is to fabricate detectors with performance being driven by signal fluctuation noise (in the UV and VIS bands) and background radiation noise (BLIP in the IR band).

This study first explains the general difference in the physical properties of bulk and LDS materials used for the photon detectors’ fabrication. It then outlines the impact of basic material properties (especially absorption coefficients, carrier lifetimes, and carrier mobilities) on the photodetectors’ performance for both standard devices and emerging next-generation photodetectors. Among the considered LDS materials, the most important are Ga_2_O_3_, TMDs, perovskites, and QDs. Particular attention is paid to the detectors’ performance limitations. Finally, this paper also attempts to predict the place of LDSs in the large photodetector family in the longer term.

## 2. General Classification of Photon Detectors

Similar to standard devices, LDS photodetectors may be divided into two groups: photon and thermal detectors. The second is related to thermal mechanisms, including, e.g., bolometric and photothermoelectric (PTE) effects. In this paper, only photon detectors are considered.

Depending on the interaction mechanism, the photon detector family is sub-divided into further groups: intrinsic, extrinsic, photoemissive (Schottky barriers), and quantum well detectors [2]. The selected photodetectors are described in sketches in Table 1 [6].

Considering the different types of photon detectors described above, it can be concluded that they can be broadly divided into two groups: photoconductive and photovoltaic. Their characteristic parameters are summarized in Table 2.

FET/hybrid designs, a current research goal for LDS photodetectors, allow one to improve responsivity; however, the considerable number of those detectors exhibits a limited LDR caused by the charge relaxation time saturating the possible photoexcitation states, causing the responsivity decrease versus light power. The applications require high-performance photodetectors exhibiting a wide LDR (linear photocurrent dependence on the light power before saturation of absorption, *I_ph_* ∝ *P^α^*, where *α* is close to 1). In terms of the LDS detectors (to include 2D material-based photodetectors) a complicated carrier generation–recombination and trapping mechanisms drive exponent 0 < *α* < 1, driving the detector’s responsivity according to the equation *R* = *I_ph_*/*P,* leading to *R* ∝ *P^−(^*^1−*α*)^. The net photocurrent and responsivity nonlinear dependence versus light power is presented in Figure 1a. Similar behavior is observed for photoelectric gain being strongly influenced by traps [refer to Figure 1b]. If radiation power increases, the carriers are progressively captured in filling traps, which decreases carrier lifetime and photoelectric gain, while for the low-radiation power range, the sensitivity is not affected because of the high density of trap states. Generally, sensitivity being measured for different radiation powers is not used for comparing the detectors’ performance. To measure the current responsivity, it is essential to implement the light power density < 1 mW/cm^2^ by several orders of magnitude [7].

## 3. Fundamental Detectivity Limits

The photon detectors reach the most favorable conditions when the intrinsic noise is low in comparison to the radiation-induced noise [8,9,10]. The photon noise level is not related to the deficiencies in the device design or coupled electronics but is connected to the detection mechanism being determined by the nature of the electromagnetic radiation. The photons incident on the device are built of two parts, coming from the target and scene. There are two detector fundamental performance maximum values: the background fluctuation limit, known as the BLIP (background-limited infrared photodetector), and the signal fluctuation limit (SFL). Both SFL and BLIP detectivities defining those ultimate limits are given in Table 2.

Figure 2 presents the ultimate detectivity reported for selected photon detectors within a 0.1–20 μm wavelength range for a 300 K scene temperature and 2π field of view (FOV). As shown, the SFL and BLIP curve crossing point is located ~1.2 μm, and for <1.2 μm, the device operates under SFL, where the *D** dependence on wavelength is weak; for >1.2 μm, BLIP dominates and *D** dependence is strong, resulting from intense increase in the scene radiation influence.

As presented in Figure 2, in the UV range, the highest *D** above the SFL is given for LDS photodetectors, primarily for AlGaN HEMT and Ga_2_O_3_ FETs. In terms of typical detectors, AlGaN photodiodes exhibit the highest *D** at 260 nm; however, to reach that high *D**, it is essential to use filters to suppress solar-irradiance leakage [11]. The utmost *D** for LDS photodetectors is reported in Refs. [17,18,19] (ZnO nanowire detectors reach the SFL [17]). The CdTe nanocrystals reach *D**~5 × 10^17^ cmHz^1/2^/W, reported as the highest recorded in VIS and IR detectors at 300 K (actually, in NIR~850 nm), 4–5 orders of magnitude higher than commercially available crystalline Si photodiodes [19]. It must be stressed that it was not reported what filters were used for the performance measurements (that record-breaking performances could not be achieved—it is believed those *D** is rather overestimated [17,19]).

Also, in the IR range, the highest *D** (data exceeding BLIP were published) was reported for LDS (primarily 2D TMDs) photodetectors [24,26]. In terms of the LDS photodetector *D** estimation, the contribution of the different types of noise should be assumed to include shot, generation–recombination (g-r), photon, thermal and 1/*f* noises; however, there are many reports not considering the photogain effect’s influence on the shot and g-r noises, mainly for devices (at 300 K) exhibiting a low response time caused by long carrier lifetimes (typical for 2D material-based hybrid photodetectors/phototransistors). Assuming incorrect expression for the shot noise ($I_{sh}=\sqrt{2qI∆f}$ rather than of $I_{sh}=\sqrt{2qgI∆f}$) causes a false improvement (record-breaking performances) in the signal-to-noise ratio (SNR) by ×$\sqrt{g}$ in relation to the SNR at the device input. A similar dependence on $\sqrt{g}$ occurs for generation–recombination noise since $I_{gr}=\sqrt{4qI_{d}g∆f/\left(1+\omega^{2}\tau^{2}\right)}$ (the g-r noise is frequency-related, *ω*). The shot/g-r noises’ estimated error increases for higher *g* and is principally important for detectors exhibiting high gain. Figure 3 presents *D** versus *g* for selected UV photodetectors. The highest *D** to include exceeding SFL is marked for Ga_2_O_3_ FET phototransistors and LDS photodetectors. Theoretical predictions for SFL within a wavelength range of 250–360 nm are also shown.

There are several reports stressing cases where 2D layered photodetectors’ performance is overestimated by the wrong characterization procedures, including the following [2,3,4,5]:Inaccurate noise assessment.Wrong assessment of the radiation power density and device active area.Noise and responsivity contrary bandwidth.Nonlinearity in light response and in the electrical conduction.

Based on these issues, the LDS detectors require appropriate characterization methods being compatible with those used for standard bulk detectors.

Figure 4 summarizes the spectral responsivity for LDS photodetectors, reaching up to ~10^10^ A/W and gain > 10^9^ electrons per photon for *λ* < 2 μm (record *g*~10^10^ was reported) [18,19,20,28,29,32,36,37,38,39,40,41]. Gain, *g*~1 for the photodiodes is caused by the carriers’ separation by the electric field in the depletion layer. Figure 4 also presents record current responsivity demonstrated in the IR~10^3^ A/W for WS_2_/HfS_2_ heterojunctions [39] (gain contribution) compared with ~10 A/W for HgCdTe photodiodes (*g*~1).

## 4. Ga_2_O_3_ Ultraviolet Photodetectors

To fabricate a UV detector, the Si and selected III–V materials, GaAsP, GaP, were originally implemented; however, costly optical filters (needed for a substantial reduction in the signal reaching the detector) are essential to adjust the device to the proper wavelength. To circumvent the requirements for attenuation filters, wide-bandgap semiconductors including SiC, GaN, or AlGaN are implemented. Despite that, Si maintains a strong position. Table 3 provides the parameters of important compounds used for commercially available UV detectors’ fabrication.

Recently, huge progress in the Ga_2_O_3_ UV photodetectors’ performance has been observed. Looking back to history, Ga_2_O_3_ was found in the 1950s and developed as a probable replacement for various powerful materials for device applications to include semiconductor (FET) and optoelectronics devices. The atomic arrangement determines seven distinct Ga_2_O_3_ crystal structures, and, among them, β-Ga_2_O_3_ (*E_g_* = 4.9 eV) is the most stable and broadly implemented for solar-blind photodetectors (SBDs) [42].

The Ga_2_O_3_ photodetectors are fabricated based on both binary and ternary alloy single crystals, epitaxial films, and nanostructures. P-type doping struggles to reach what is related to the holes being trapped by local lattice structures, resulting in a huge effective mass and limited mobility; however, more significantly, 6″ Ga_2_O_3_ single crystals and epitaxial films are obtainable [43]. What is more, the Ga_2_O_3_ can be also mechanically exfoliated and promptly transferred to a substrate (FET phototransistors’ fabrication).

The Ga_2_O_3_ photodetectors’ performance—current responsivity, response time, detectivity, and gain (carrier trapping and persistent photoconductivity)—has been found to be spread over a broad range due to various device structures and properties of the crystal/material phases (see Figure 3). Figure 5 presents the β-Ga_2_O_3_ photodiode current responsivity compared to the wide-bandgap GaN- and SiC-based commercial devices. The β-Ga_2_O_3_-based photodiode performance is higher compared to the commercial devices operating in the UV-C range of <250 nm. The vertical barrier Gr/MoS_2_/h-BN 2D FET heterostructure detector also exhibits high responsivity in the UV-A band [44].

## 5. Perovskite Photodetectors

The term “perovskite” was adopted when the mineral CaTiO_3_ was identified by Gustav Rose in 1839 in the Urals, named in honor of L.A. Perovski (Russian mineralogist), who conducted extensive research on its structure. Since then, the term perovskite has been expanded to include all compounds that have an identical or similar crystal structure to CaTiO_3_.

Perovskites are promising materials for upcoming high-performance applications to include typical devices to include the following: photodetectors, lasers, solar cells, light-emitting diodes, and disruptive technologies related to the pressure-induced emission and artificial synapses/memristors–neuromorphic devices. Perovskite-based devices have elicited broad attention caused by long carrier diffusion lengths, high absorption coefficients, high defect acceptance, and, finally, direct bandgap. Based on the fact that perovskite devices have exhibited remarkable performance to be used in thorough interpretation of chemistry and solid state/photo physics, the progress is principally initiated by the innovative development of next-generation solid-state perovskite solar cells.

By changing the elemental species in the perovskite materials, the optical and electrical properties and stability of perovskites can be significantly changed. Unlike conventional semiconductor materials, in which defects act as trap states located between the top of the valence band (VB) and the bottom of the conduction band (CB), the orbitals in perovskites are located inside or near the edges of the VB and CB bands. This makes perovskites highly tolerant to defects. This benefit is particularly evident in flexible LEDs, which must withstand various mechanical deformations.

Perovskites are fabricated using simple and flexible low-energy preparation methods. The metal halide perovskite ionic feature offers a low-cost solution precursor approach via “wet chemistry”, similar to organic semiconductors and colloidal quantum dots (CQDs).

The critical problem of lead perovskite with organic cations that needs to be solved is thermal and structural instability. The 3D MA-based perovskites (MA—methylammonium) were found to be instable due to the hazardous MA cations. Pb-toxicity MAPbI_3_ is susceptible to heat, moisture, oxygen, and light, casing material degradation, while FA-based perovskites (FA—formamidinium) were found to be much more thermally stable than MA. Generally, inorganic perovskites (CsPbI_3_) exhibit better thermal stability. Several elements have been implemented to replace Pb, among them Sn, Ge, Bi, and Cu; but devices manufactured from these materials are less efficient.

Motivated by atomic thin 2D materials’ outstanding performance, 2D perovskite-based materials exhibit significant prospects for functional applications. In addition, 2D perovskite materials reach comparable optoelectronics performance for 3D bulk crystals while exhibiting better humidity tolerance, which consequently contributes to the better luminescence efficiency and devices’ longer-term stability.

The perovskite active layers combine several advantages (seen for well-known PV technologies) to include the following: high-efficiency (reported for bulk Si PV), lightweight/flexibility (CdTe, GaAs, CIGS inorganic thin film PVs), scalable low-temperature solution-based fabrication and wavelength tuning (dye-sensitized, organic and QD-based thin-film PVs) [45]. The main optoelectronics parameters (energy loss, absorption coefficient, power per weight and electronic properties) of perovskite solar cells compared to other thin-film PVs are presented in Figure 6.

Table 4 presents the unique parameters of perovskite materials [47]. The high efficiency of PV devices is driven by the large electron mobility ~200 cm^2^/Vs connected to a long diffusion length > 1 μm, absorption coefficients ~10^5^ cm^−1^ (caused by s-p antibonding coupling), and low exciton binding energy < 10 meV, allowing the excited carriers to be freely transported [48].

Figure 7 presents the responsivity and detectivity for the selected perovskite photodetectors (different material compositions, morphologies, and device configurations) [49]. The photodetectors operate in the VIS and NIR range. Normally, photodiodes need a reasonably low operating bias to reach a large detectivity in comparison with photoconductors. Higher current responsivities (caused by the influence of the photogating effect) are presented for hybrid phototransistors and photoconductors. The best perovskite photodetectors’ performance reaches ~10^4^ A/W (current responsivity), ~10^14^ Jones (detectivity), high LDR, and nanoseconds’ range response time. This performance shows a strong alternative and competition for Si-based devices in sensing and imaging applications.

Figure 8 presents the responsivity and detectivity for perovskite photodetectors, where a large discrepancy is visible, mainly for hybrid photodetectors [see Figure 8a]. That trend is caused by undeveloped device technology and the photogating contribution. It is believed that the reported performance for the FAPbI_3_QD/VAGA photodetector [vertically aligned graphene arrays (VAGAs) integrated with FAPbI_3_ QDs], >10^15^ Jones at 1.55 μm is overestimated [50].

In summary, perovskite materials have gained research attention in the last decade due to excellent processability and excellent carrier transport capabilities like typical semiconductors. The synergetic perovskites are incorporated into numerous device applications, including photodetectors, thin-films solar cells, light-emitting diodes, lasers, and transistors. However, from an application point of view, several issues remain unresolved:Stability—Generally, inorganic perovskites, due to their ionic nature, are found to be not fully stable. In a humid environment, perovskites tend to drastically change their crystallographic structure and even their composition, causing irreversible damage to the material. Effective housing is required to fully protect the device and provide long-term stability under extremely severe operating conditions.Toxicity—So far, Pb-based perovskites have been found to reach to the best performance; however, Pb-free perovskites have been gradually introduced, e.g., Bi-based perovskites and double perovskites.Miniaturization—This is principally important in imaging. The perovskite fabrication methods significantly differ from silicon technology to include conventional etching/lithography techniques (essential to develop a mature technology comparable with silicon). In addition, there is a need to develop ROICs for perovskite arrays, where specific designs are required due to the high sensor’s RC.Immature perovskite quantum dots (PQDs) technology is behind typical QDs (immediate and proper solution to allow PQDs to open up a new era, especially in the field of solar cells).

To sum up, it is believed that further expansion and development of the perovskite materials family within optoelectronic applications still lie ahead of us.

## 6. Infrared Photodetectors: LDS vs. Standard Detectors

Generally, the IR detector material is chosen based mainly on the wavelength, operating temperature, and performance. According to the data presented in Figure 9, presently, microbolometers, III–Vs/InSb, HgCdTe and Si:As blocked impurity band (BIB) are the most common materials for IR detectors’ fabrication.

InGaAs lattice matched to InP were found to be the proper material for NIR (1.0–1.7 μm). MCT, primarily in PV configurations, covers the 0.7–20 μm range. InAs/GaSb type-II superlattices (T2SLs) have been found to be a replacement for MCT. Silicon impurity-doped (Sb, As, Ga) BIB detectors, operating at 10 K, exhibit a cut-off wavelength (*λ_c_*) within the 16–30 μm range. Ge:Ga PCs have been found to exhibit the highest performance for low-background photon detectors for 40–120 μm operating at ~2 K. The selected 300 K semiconductor material parameters, important from the photodetector’s design point of view, are presented in Table 5.

Recently, thorough attempts have been undertaken to limit the IR systems’ cost and increase the operating temperature by reducing SWaP. According to our knowledge, “*there is no fundamental obstacle to obtaining room temperature operation of photon detectors with background-limited performance even in reduced fields of view*” [51]. Assuming the present state of the art in IR detector technology, it may be concluded that those goals are nearly reached. It is believed that the significance of photon IR detectors operating at 300 K (HOT—high operating temperature) will increase in the near future [52].

The BLIP condition is driven by Law 19, typical for P-i-N MCT photodiodes. Moreover, except for the standard semiconductor, which includes MCT and III–V T2SLs, the 2D materials and colloidal quantum dots (CQDs) should be accounted for in the materials (proper physical properties) required for IR HOT devices. In the following section, new-generation “*third wave*” materials will be considered in detail.

### 6.1. Fundamental Material Properties

Even though hybrid graphene-based phototransistors exhibit very high responsivity, their applications are being constrained by the electronic readout circuits, power usage, and noise, being driven by zero bandgap, leading to a high dark current. The finding of the new layered materials (LMs) exhibiting direct bandgap energies within the VIS-IR range initiated a new era for photodetector fabrication [1]. There are hundreds of LMs that are stable down to the constituent monolayers. The LM properties/parameters have been found to be complementary and comparable to graphene and were reported in many review papers. The low-bandgap 2D transition metal dichalcogenides (TMDs with Ni, Pd, Pt metals) exhibiting a bandgap energy of ~0–0.25 eV could be an innovative approach to reach 300 K and long-wavelength infrared (LWIR) conditions [53,54,55].

The quantum confinement and surface effects allow 2D TMDs to exhibit layer-dependent capabilities, considerably differing from bulk crystals. The TMD energy band structure transfers from low indirect to a wide direct bandgap versus material thickness (transiting from the bulk to LM). That allows TMDs to detect radiation with a wide range of wavelengths by varying the number of layers (bandgap adjusted by quantum confinement effects) [56,57]. In addition, the optical/electronic parameters may be greatly influenced by the strains. The TMDs that include MoS_2_, WS_2_, and MoSe_2_ demonstrate even larger absorption in both VIS and NIR than graphene.

A comparison of the TMDs’ main optoelectronic properties, including absorption coefficients, carrier lifetimes, and carrier mobilities, with typical materials used for IR photodetectors’ fabrication, is presented in Figure 10. The absorption coefficients for TMDs with 1–2 eV bandgap normally reach ~10^5^–10^6^ cm^−1^ [refer to Figure 10a], meaning that more than 95% of the sunlight is absorbed by sub-micrometer-layer thickness, which can be clarified by dipole transitions between localized d-states and excitonic coupling. The data presented in Figure 10a allow us to draw the following conclusions:MCT (absorption coefficients, *α* < 10^4^ cm^−1^, *E_g_* ≈ 0.1–0.3 eV) and for TMDs (*α* > 10^5^ cm^−1^, *E_g_* > 1 eV).TMD (theoretical *E_g_* ≈ 0.1–0.2 eV) absorption coefficient < 10^5^ cm^−1^.

**Figure 10 materials-17-04522-f010:**
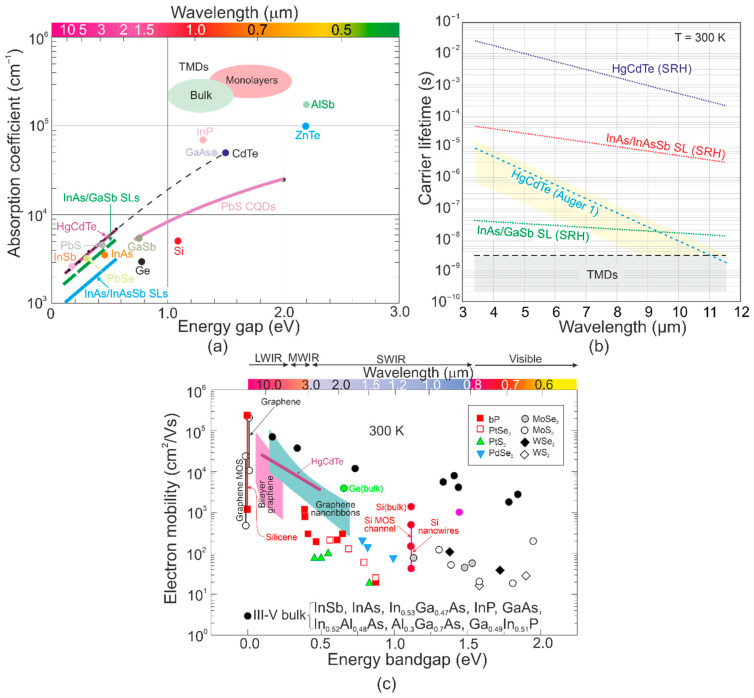
Comparison of the 300 K TMDs optical and electric parameters with typical materials used for IR photodetectors’ fabrication: (**a**) absorption coefficient versus bandgap; (**b**) minority carrier lifetimes for HgCdTe, both T2SLs InAs/GaSb and InAs/InAsSb and TMDs; (**c**) electron mobilities for LDS, selected III–V compounds and HgCdTe.

The carrier lifetime is considered to be a critical parameter determining the detector’s response time and detectivity. Generally, the carrier lifetime for 2D materials decreases versus the incident radiation power, imposing reliable lifetime measurements under low-power conditions.

Figure 10b presents the minority carrier lifetime as a function of *λ_c_* for selected materials at 300 K [58]. HgCdTe exhibits promising internal recombination mechanisms, determining the long carrier lifetime, allowing it to reach HOT conditions. The lightly doped MCT (~10^13^ cm^−3^) carrier lifetime driven by the Shockley–Read–Hall (SRH) process allows it to reach ~10 ms (MWIR) and ~0.5 ms (LWIR), respectively (extracted from the photodiode dark currents characteristics at low temperatures) [59]. Average values may be assumed for devices operating at 300 K; however, it must be stressed that the large intrinsic carrier concentration ~6 × 10^15^ cm^−3^ (MWIR) and 5 × 10^16^ cm^−3^ (LWIR) at 300 K limits the carrier lifetimes through Auger processes—Auger 1 (n-type) and Auger 7 (p-type). Intrinsic Auger 1 and Auger 7 carrier lifetimes in HgCdTe follow the relation *τ_Ai_*_7_ ≈ 6*τ_Ai_*_1_ [60].

Generally, III–V materials exhibit more SRH traps than II–VI, including HgCdTe influencing minority carrier lifetimes. II–VI compounds exhibit stronger ionic bonds than III–V counterparts, making the electron wave function around the lattice sites more compact and less susceptible to the formation of band states due to crystal imperfection. The T2SLs InAs/GaSb carrier lifetime (driven by the SRH) reaches several tens of nanoseconds (both MWIR and LWIR) and is attributed to the Ga occurrence, while “Ga-free” T2SLs InAs/InAsSb demonstrate longer lifetimes, up to several microseconds in the MWIR.

The TMDs’ carrier lifetime (reaching the nanoseconds range) is considerably lower than the bulk and T2SL typical photodiodes. The TMDs’ carrier mobility increases versus the number of layers, but the mobility is low (reaching < 250 cm^2^/Vs, limited by ripples, phonon scattering, impurity scattering, and interface scattering) and was found to be difficult to increase, meaning that 2D materials do not exhibit unique pros over the standard 3D bulk compounds (both III–V and II–VI). Figure 10c summarizes the 300 K carrier mobility of typical group-6 TMDs and PtSe_2_, PtS_2_, PdSe_2_ (noble TMDs with selected number of constituent layers), and bP on the back-gated SiO_2_ substrates. The 10^12^ cm^−2^ charge carrier density was reported and found to be dependent on the doping levels and recombination centers.

### 6.2. HgCdTe Benchmarks

The “Rule 07” metric has been universally used by the IR community to predict the HgCdTe photodiode performance for more than 17 years [61]. In addition, it could be implemented as a point of reference for possible alternate technologies, such as III–V-based barrier detectors [62,63], T2SL devices [22,63,64], QD photodetectors [65,66], and 2D material photodetectors [67].

#### 6.2.1. “Rule 07”

Tennant et al. [61] presented a formula describing the dark current density versus wavelength and temperature for the best-performing Teledyne MCR diodes and arrays [predominantly double-layer planar heterojunction (DLPH)]. It follows theoretical characteristics for an Auger-suppressed p-on-n photodiode with ~10^15^ cm^−3^ electron concentration in the absorber according to the relations [61]:(1)$$J_{dark}=8367exp\left(-\frac{1.44212q}{k\lambda_{c}T}\right)\;\text{for}\;\lambda_{c}\geq 4.635\;\mu m$$
and
(2)$$J_{dark}=8367exp\left\{-\frac{1.44212q}{k\lambda_{c}T}\left[1-0.2008{\left(\frac{4.635-\lambda_{c}}{4.635\lambda_{c}}\right)}^{0.544}\right]\right\}\;\text{for}\;\lambda_{c}<4.635\;\mu m$$
where: *λ_c_* [μm]—cut-off wavelength, *T* [K]—operating temperature, *q*—the electron charge, *k*—Boltzmann constant. “Rule 07” is valid for cut-off wavelength-operating temperature (*λ_c_T*) products within the range 400 μmK–1700 μmK and > 77 K.

#### 6.2.2. “Rule 22”

Given the present status of HgCdTe technology, “Rule 07” was found not to be a proper method to assess the ultimate performance of HgCdTe photodiodes. Lately developed, “Rule 22” perfectly describes the non-Auger-limited dark current of the highest-performing HgCdTe photodiodes [68]. That simple empirical equation covers a wavelength range of *λ_c_* = 1.6–17 μm and *T* = 20–330 K. As presented in Figure 11, for 1/*λ_c_ T* < 0.0025), the p-on-n device is restricted by Auger-1 diffusion currents, and 0.0025 < 1/*λ_c_ T* < 0.005 electron trap energy, *E_T_* = 0.39*E_g_* (trap-assisted tunnelling) limits the device’s performance. In a range of 1/*λ_c_ T* > 0.005, the photodiode dark current is limited by very low 3.7 ph/cm^2^s background flux. It is believed that the further suppression of defects (traps) and scene flux will reduce the dark current for 1/*λ_c_ T* > 0.0025.

“Rule 22” is given by the following equation [68]:(3)$$J=\left(1\times 10^{7}\times \lambda^{-6.2}+70\times \lambda^{1.08}\right)\times \left\{exp\left(-1.04\times \frac{1.24}{\lambda }\times \frac{q}{kT}\right)\right.\;\left.+2\times 10^{-10}\times exp\left(-0.39\times \frac{1.24}{\lambda }\times \frac{q}{kT}\right)\right\}\;+1.5\times 10^{-21}\times \lambda^{2}T.$$

It was found to be in proper agreement with the measured results for mesa and planar DLPHs deposited by MBE, where the wide-bandgap “cap” layer was grown on the absorber, through which As is implanted to create the junction, leaving the junction perimeter intersecting the surface in the wide-bandgap “cap” region [refer to Figure 12a,b]. The electric fields dropping at interfaces block minority carriers, reducing the surface recombination contribution [refer to Figure 12c]. The surface recombination may also be limited by proper passivation. The absorber thickness ranges from 4 μm (NIR) to 15 μm (very-long-wave infrared: VLWIR), while n-type doping reaches 0.5–2.0 × 10^15^ cm^−3^. The presented data cover the saturation dark current for −500 mV < *V_b_* < −50 mV. More information on DLHJ HgCdTe technology can be found in Ref. [69].

#### 6.2.3. “Law 19”

The “Rule 22” benchmark is related to the low-doped mid i-region (refer to Table 1); however, for satisfactorily long HgCdTe SRH carrier lifetimes (experimentally confirmed at doping levels <5 × 10^13^ cm^−3^), the P-i-N MCT photodiode intrinsic current is reduced, and the performance is suppressed by scene radiation [58,59,70]. The intrinsic photodiode current is lower than that driven by both “Rule 07” and “Rule 22”, depending on the *λ_c_* and *T*. That condition imposes the new “Law 19” benchmark to assess the best MCT photodiode performance in the LWIR.

Assuming that the P-i-N photodiode’s i-region is completely depleted, the depletion current can be calculated using the following relation:(4)$$J_{GR}=\frac{qn_{i}W}{\tau_{SRH}},$$
where: *W*—depletion region thickness depending on voltage (*V*) and doping (*N_d_*) and may be estimated by ${W=\left[2\varepsilon \varepsilon_{0}\left(E_{g}+V\right)/qN_{d}\right]}^{1/2}$, where: *E_g_*—bandgap energy, *ε*—dielectric constant, and *ε*_0_—vacuum permittivity. The Hg_1−x_Cd_x_Te (composition and temperature dependence) may be estimated using Hansen’s formula [71], with the dielectric constant (ε) according to the relation $20.5-15.5x+5.7x^{2}$.

When the carrier lifetimes are suitably long, the current is reduced, and the device’s performance is restricted by the scene radiation—BLIP condition according to the relation:(5)$$J_{BLIP}=\eta q\phi_{B},$$
where: *q*—electric charge, *η*—quantum efficiency, *ϕ_B_*—net background photon flux density.
(6)$$\phi_{B}=\sin^{2}\left(\theta /2\right)\int_{0}^{\lambda_{c}}\frac{2\pi c}{\lambda^{4}\left[exp\left(hc/\lambda kT_{B}\right)-1\right]}d\lambda ,$$
where: *θ*—cone angle, *c*—light velocity, *λ_c_*—cut-off wavelength, *k*—Boltzmann, *h*—Planck constant, and *T_B_*—background temperature. Assuming an ideal *η =* 1, relation (5) allows one to extract the analytical expression for “Law 19” [59]:(7)$$J_{rad}=qC_{1}\left(a^{2}+2a+2\right)exp\left(-a\right),$$
where:(8)$$C_{1}=1.7\times 10^{18}{\left(T/300\right)}^{3},$$
(9)$$a=48\times \left(300/\lambda_{c}T\right),$$

*T*—detector temperature.

The dark current components should be considered as limiting factors, representing shot noise sources, deteriorating the expected signal-to-noise ratio (SNR). Depending on the device’s design and the carrier recombination lifetimes, the dark current contribution can be suppressed. The device’s detectivity restricted by the scene noise may be expressed by a relation given in Table 2.

Below the experimental dark current densities for MCT, T2SLs, InGaAs, InSb, 2D material, and CQD-based photodetectors were collected to compare with “Law 19”—BLIP driven at 2π FOV and 300 K background depicted by a red line in Figure 13. The figure presents the dark current densities for selected types of LWIR photodetectors versus the reverse of the *λ_c_T* product. The highlighted data were extracted from many reported papers. For 2D material-based photodetectors and CQDs, the dark current density is assessed against the results reported in papers.

As presented in Figure 13, the typical IR detectors determining the state-of-the-art devices demonstrate performance below the BLIP limit, with the dark current exceeding the ultimate scene-limited level at 300 K for 2π FOV (red line). The BLIP conditions contribute primarily to current density (low 1/*λ_c_T*) for photodiodes operating in LWIR and HOT conditions. The lower current density of T2SLs interband quantum cascade photodetectors (III–V IB QCIP) compared to MCT photodiodes in the LWIR region at 300 K (low 1/*λ_c_T*) should be underlined. Conversely, the lowest dark current densities estimated for 2D material-based photodetectors based on the reported papers located below BLIP (red line) show that the border (defined by the BLIP) has been exceeded, meaning that some results reported on the 2D material-based photodetectors indicate the device’s performance overestimate.

### 6.3. Figure of Merit for Infrared Detector Materials

Table 2 defines the fundamental figure of merit for photoconductors and photodiodes. In this section, we take a closer look at the most key parameters of 2D material-based photodetectors compared with typical commercial detectors dominating the global market. More attention will be paid to quantum efficiency, detectivity, and gain.

#### 6.3.1. Quantum Efficiency

Quantum efficiency may be considered as external, *η_ext_*, and internal, *η_int_*. *η_int_* defines the ratio of collected carriers to the absorbed photons, but *η_ext_* shows the incident photons. If we assume that the gain, *g*, and carrier transfer efficiency are equal to 1, then the *η_int_* and *η_ext_* will be <1; however, *η_int_* is higher than *η_ext_* for some wavelengths and is given by the following relation:(10)$$\eta_{ext}=g\eta_{int}=\beta g\left(1-r\right)\left(1-e^{-\alpha x}\right),$$
where: *β*—carrier transfer efficiency, *r*—reflectivity, and *x*—position within the material. Equation (10) indicates that the photogain has a decisive influence on the *η_ext_*.

Figure 14 compares the thickness-normalized external quantum efficiency (thickness is normalized at 1 nm) between 2D materials and traditional epitaxial layers [72]. The majority of the 2D materials show a better photoelectric conversion capability than the thin film of the narrow-bandgap HgCdTe, which is related to the 2D materials’ higher absorption coefficients [refer to Figure 10a] and photoelectric gain contribution; however, the ultra-thin 2D materials with thicknesses ranging from a monolayer (few angstroms) to tens of nanometers exhibit much lower internal quantum efficiency compared to the typical bulk epitaxial layers.

#### 6.3.2. Detectivity

Piotrowski and Rogalski reported that IR detectors’ performance is limited by the probabilistic nature of the generation–recombination mechanism and may be calculated according to the following formula [73]:(11)$$D^{*}=k\frac{\lambda }{hc}{\left(\frac{\alpha }{G_{th}}\right)}^{1/2},$$
where: *h*—Planck’s constant, *λ*—wavelength, *α*—absorption coefficient, *c*—light speed, and *Gth* (in cm^–3^s^–1^)—thermal generation in the absorber, *k*—coefficient presenting radiation coupling to detector: microcavities, plasmonic structures or antireflection coating. The *α*/*Gth* ratio is considered to be the IR material figure of merit and may be used to predict the ultimate performance and to select materials for the detector’s absorber. *Gth* presents the rate at which a perturbed carrier system returns to equilibrium. In terms of the reverse-biased photodiode, it describes the rate at which carriers are generated within the diffusion length of the junction and, as reported by Kinch, may be expressed by [60]:(12)$$G_{th}=\frac{N_{min}}{\tau },$$
where: *N_min_*—minority carrier density, *τ*—minority carrier lifetime. Since
(13)$$N_{min}N_{maj}=n_{i}^{2},$$

The minority carriers *Gth* (holes in n-type MCT active layer) may be calculated according to the following expression:(14)$$G_{th}=\frac{N_{min}}{\tau }=\frac{n_{i}^{2}}{N_{maj}\tau },$$
where: *n_i_*—intrinsic carrier concentration, *N_maj_*—majority carrier density. Finally, the photodiode *D** can be estimated by the following relation:(15)$$D^{*}\propto {\left(\frac{\alpha }{G_{th}}\right)}^{1/2}={\left(\frac{\alpha N_{maj}\tau }{n_{i}^{2}}\right)}^{1/2}=\left(\frac{\sqrt{N_{maj}}}{n_{i}}\right)\sqrt{\alpha \tau },$$
meaning that *D** is proportional to the $\sqrt{\alpha \tau }$ product.

Table 6 shows the assessed $\sqrt{\alpha \tau }$ for HgCdTe and both T2SLs InAs/GaSb and “Ga-free” InAs/InAsSb. The highest reported values of SRH carrier lifetimes (300 K) may be reached only by the proper detector design, where the intrinsically generated carriers are either suppressed to the doping level or to complete depletion of the absorber (e.g., HgCdTe P-i-N). Between the T2SLs InAs/GaSb and “Ga-free”, the difference in the $\sqrt{\alpha \tau }$ was estimated to be within two orders of magnitude for MWIR and one order of magnitude for LWIR. T2SLs exhibit a lower $\sqrt{\alpha \tau }$ than HgCdTe, being related to much longer HgCdTe carrier lifetimes.

The 2D materials’ carrier lifetime was reported at the level of nanoseconds and to be around five orders of magnitude lower than for HgCdTe [refer to Figure 10b]. The exciton recombination mechanism was found to be predominant for MoS_2_, MoSe_2,_ and WSe_2_ (TMDs) [75,76]. Analyzing the measured results for LWIR MCT (*α* = 2.2 × 10^3^ cm^−1^, *τ* = 0.5 ms) and TMDs (*α* = 2 × 10^5^ cm^−1^, *τ* = 1 ns), it is essential to analyze $\sqrt{\alpha \tau }$ products for 2D materials. For HgCdTe, $\sqrt{\alpha \tau }$ was estimated at the level 1.05 (s/cm)^1/2^, but for TMDs, 1.4 × 10^−2^ (s/cm)^1/2^ was reported as being two orders of magnitude lower and close to that predicted for T2SLs InAs/InAsSb. It must be stressed that the $\sqrt{\alpha \tau }$ product was compared for materials with considerably different energy bandgaps. The TMD absorption coefficient was assumed for the ~1 eV bandgap, while, for MCT, it was ~0.1 eV. Theoretically, for a 2D material with a bandgap (~0.1 eV), the absorption coefficient should be lower (nearly one order of magnitude), resulting in a lower $\sqrt{\alpha \tau }$ than HgCdTe.

Figure 15 presents the reported *D** versus wavelength for selected single-pixel IR photodetectors operating at 300 K, including LDS, 2D material-based devices, and CQDs. The performance of typical photodetectors based on the HgCdTe, T2SLs, PbS, PbSe, InGaAs was estimated to be below the BLIP limit. The T2SL-based IB QCIPs’ performance was found to be comparable to MCT photodiodes. However, the complex design with multiple interfaces and strained thin layers makes the technology difficult to use and expensive. The highest *D** in the IR range was reported for LDS (bPAs and TMDs) photodetectors.

Figure 16 complements the results presented in Figure 15 reported for LDS photodetectors with a theoretically calculated performance benchmark for HgCdTe (“Rule 07”, “Law 19”, and “Law 22”). In addition, the MCT photodiode performance trend lines and 2D material-based detectors are depicted. It was found that the highest *D** for LDS photodetectors within 2–12 μm is higher than for typical IR detectors fabricated based on InGaAs, PbS, PbSe, HgCdTe, and T2SLs.

The record *D**~5 × 10^11^ cmHz^1/2^/W at 300 K and 5–8 μm for WS_2_/HfS_2_ is reported in Ref. [39]; however, *D** is higher than BLIP for the 300 K scene and 2π FOV. The corrected *D** (indicated by red arrows) is the best performance reported to date for IR photodetectors operating at 300 K in a wavelength range of 5–7 μm [24]. The most probable reason for the highest *D** overestimate shown in both Figure 15 and Figure 16 is internal gain and noise miscalculation. A similar overestimate of *D** by at least an order of magnitude is reported for the Bi_2_O_2_Te 2D thin-film photodetector (wavelength of 2.4 μm) [26].

**Figure 16 materials-17-04522-f016:**
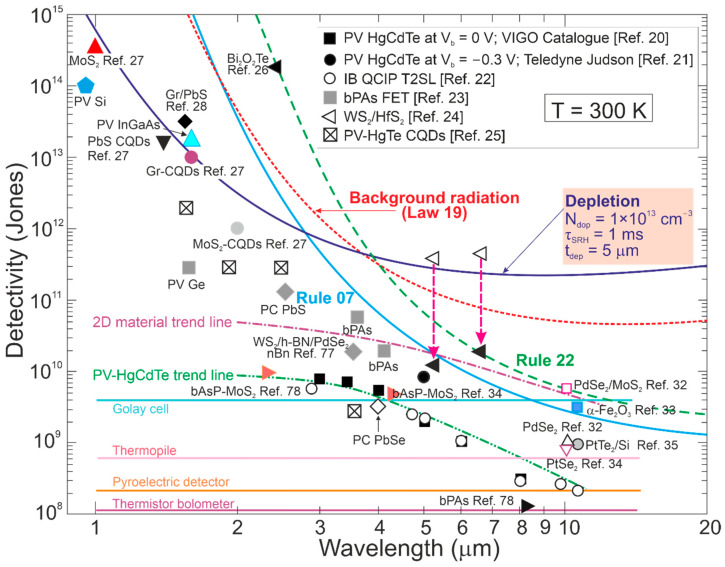
Detectivity for available in the market IR detectors operating at 300 K (PV-Ge, PV-InGaAs, PV-Si, PC-PbSe, PC-PbS, PV-MCT [20,21]). Measured data shown for T2SLs IB QCIPs [22], selected 2D materials [23,24,26,27,28,32,33,34,35,77,78] and CQD [25] photodetectors. Theoretical simulations (curve labelled as “Depletion”) presented for P-i-N HOT MCT photodiodes with thickness *t*~5 μm, *τ_SRH_* = 1 ms and absorber doping ~1 × 10^13^ cm^−3^. For comparison reasons the “Rule 07” and “Law 22” curves and typical thermal detectors (pyroelectric detector, thermopile and Golay cell, thermistor bolometer) are shown. FET—field-effect transistors, PV—photodiode, PC—photoconductor.

Figure 16 depicts the present HOT photodetectors’ performance and indicates possible progress in the future. As shown by “Law 19”, the detectivity of P-i-N MCT photodiodes (absorber thickness 5 μm, carrier lifetime in the depleted i-region *τ_SRH_* = 1 ms, doping 1 × 10^13^ cm^−3^) operating at 300 K, *λ* > 4 μm, is limited by scene radiation (reaching *D** > 10^10^ cmHz^1/2^/W) and may be increased by more than one order of magnitude in comparison with “Rule 07” in LWIR. Among the compounds used for HOT LWIR photodetectors’ fabrication, only MCT can reach the BLIP conditions (high SRH carrier lifetime ~1 ms and doping level ~10^13^ cm^−3^).

#### 6.3.3. Gain

The gain influence on the photodetectors’ performance (shot/generation–recombination noises, responsivity) is explained in Table 2 and clarified in Section 2. Two-dimensional-material-based photodetectors, primarily hybrid devices, allow one to increase the responsivity; however, most detectors demonstrate a limited LDR caused by the charge relaxation time saturating the available photoexcitation states, causing a responsivity decrease versus incident light (refer to Figure 17). In addition, the gain versus excitation intensity in comparison with the theory (solid line) for the hybrid single-layer graphene/ZnO QD-based device is presented, reaching a gain of ~10^7^ [79].

The significance of that effect is the fact that high-sensitivity detectors exhibit a low response rate, observed over a broad spectral range and confirmed by the measured results shown in Figure 18 [VIS to LWIR]. As the detector’s *λ_c_* increases, the response rate rises. Response times up to a few seconds were reported for photodetectors in VIS. It is difficult to reach both a fast response and high responsivity at the same time.

### 6.4. Two-Dimensional Material Image Sensing

Presently, single/point detectors are naturally implemented to show 2D imaging devices in NIR and SWIR; however, a point detector cannot meet the functional requirement for imaging. The first 2D material sensors operating in the VIS and SWIR are presented in [82]. The vertical, dual-band GaSe/GaSb linear array (16 × 1) was deposited by MBE in 2017 [83]. In addition, the 388 × 288 array based on the 20 μm pixel graphene/CQD hybrid single elements and CMO-based integrated device operating within UV-VIS-SWIR from 300 to 2000 nm was developed (see Figure 19) [84]. Image sensor pixels exhibit very high gain ~10^7^ and responsivity > 10^7^ A/W. The last array design has disadvantages. The pixel size is large (20 μm), limiting spatial resolution in VIS and SWIR compared to the commercial 1 μm pixel size CMOS operating in VIS. An array’s operability is assessed at the level of 95%. The spatial noise (fixed-pattern noise) strongly impacts the array’s noise-equivalent irradiance, mainly for hybrid pixels (graphene-QDs). The QDs’ nonuniformity restricts the array’s performance [85].

For future IR imaging, small-pixel detector arrays are needed; however, the high-performance arrays need high-quality/uniformity 2D material films [86,87]. The technology development in terms of high quality, homogeneous film, and wafer scale is essential and urgent.

### 6.5. Summary

Summing up, similar to graphene-based photodetectors, the 2D material IR detectors’ performance is limited by the tradeoff among the ultrafast response time, broadband operation, and high responsivity. Generally, the ultra-thin structure and insufficient absorption lead to low QE and *D**. To date, the highest IR detector performance has been demonstrated for 2D TMD materials, exhibiting strong light–matter interactions and high absorption coefficients. Oppositely, the carrier lifetimes and carrier mobilities are lower than for HgCdTe. Despite those disadvantages, the highest *D** for single-pixel devices was reported for 2D material-based photodetectors in the LWIR range and 300 K, being partly connected with the parameter overestimate. The 2D material photodetectors’ development requires properly defined characterization methods that are consistent with those used for typical bulk photodetectors. Further research is required to explain the factors contributing to the extremely high *D**.

The present 2D material-based devices’ progress is mainly related to the compounds exfoliated from bulk-layered crystals, exhibiting very low reproducibility, scalability, and yields, while FPA production needs uniform large-scale materials. The standard technologies, e.g., CVD or transfer methods, allow us to deposit small-micrometer-scale sample sizes. In the past decade, research results on the wafer-scale high-quality TMD materials’ growth have been reported, including Refs. [88,89]. Lately, the International Roadmap for Devices and Systems (IRDS) declared that TMDs exhibit the potential to replace Si in transistors and other applications beyond CMOS in 2028 [90]; however, the main drawback preventing full commercialization is the difficulty in large monocrystalline wafers’ fabrication.

The cost-effectiveness and attractiveness in the global market of the industrial production of arrays, exhibiting spatial uniformity, high operability, scalability, temporal stability, affordability, and producibility, were found to be important aspects. However, all these aspects are at an early stage of development and fabrication capabilities.

A comparison of the basic material parameters indicates that 2D materials are very unlikely to replace and reach the HgCdTe performance level in LWIR HOT devices within ten years because the mature HgCdTe technology ensures reaching BLIP conditions for HOT LWIR photodiodes (operating in a range above 4 μm). It must be underlined that, after several decades of research, development, and national/international investments, the rank of 2D materials will be strengthened to compete with the main IR players. That tendency is already visible for electronic devices (especially transistors), where Si’s fundamental limitations at the nanometric scale prevent further progress; however, it is believed that Si may increase performance upon reaching compatibility with other materials [91].

## 7. Topological Insulator Photodetectors

The quantum Hall effect, a theory that is not explained by symmetry arguments breaking, was discovered in 1980 by Klitzing [92]. Further research on the traditional quantum Hall effect led to the discovery of the quantum spin–Hall phenomena, being induced by spin–orbit coupling observed in Bi and Te heavy ions at 300 K. The spin–orbit coupling may exchange the external magnetic field for the quantum spin–Hall effect [93]. The compounds demonstrating the quantum spin–Hall effect are termed as topological insulators (TIs), having been the main research subjects in condensed-matter physics in the last decade. TIs provide wide potential applications (mechanical, electronic and optical properties), including IR photodetectors, being allowed by the presence of Dirac fermions in topologically protected surface states.

### 7.1. Fundamental Properties of Topological Insulators

TIs are a specific type of insulator with bulk insulating bandgaps and gapless conductive states guarded by time-reversal symmetry on the surface edges, allowing electrons to move along the surface of the material [94,95,96]. TIs demonstrate a strong spin–orbit coupling phenomenon, leading to the bandgap inversion shown in Figure 20.

As presented in Figure 20a, the valence-band electrons are not allowed to move to the conduction band, which is related to the wide energy bandgap; however, strong spin–orbit coupling allows the valence “spin up” band to reach the conduction band, while the conduction “spin down” band reaches the valence band, as presented in Figure 20b, leading to the band inversion, as depicted in Figure 20c for 3D TI. The presented path allows electrons to move from the valence to the conduction band without loss, and the Fermi level is located in the bulk bandgap, being traversed by topologically protected spin-textured Dirac surface states. Two edge states with opposite spin polarizations counterpropagate, which is related to the spin-momentum-locking phenomena presented in Figure 20d for 2D TIs.

To date, many TIs have been discovered and presented, and it is challenging to distinguish their types. Factually, TIs were split into 3D and 2D materials [97,98]. In comparison to the typical semiconductors, TIs exhibit the unique surface/edge electronic states, confining and allowing electrons to move along the material’s surface/edge without being transported through the volume. Similar to typical insulators, TIs exhibit a bulk energy bandgap, but the surface and edges are gapless for 2D/3D configurations, respectively. The conducting boundary state is protected by the bulk band topology increasing the carrier mobility and reducing energy losses, which makes TIs a suitable material for broadband photodetectors.

Figure 21 shows the selected TI materials, covering a wide spectral range from the UV to FIR. The bandgap may be adjusted by doping, defects, and strain engineering. To date, 3D Bi_2_Te_3_, Bi_2_Se_3_, and Sb_2_Te_3_ have been found to be the most commonly used TI materials, and the fundamental parameters are presented in Table 7 [99]. Further, 3D Bi_2_Te_3_, Bi_2_Se_3_, and Sb_2_Te_3_ properties have been found to be similar to the V and VI main groups. In addition, lattice structures are influenced by covalent bonding and mixed with ionic bonding, where each of five atoms between the quadrupole layers resembles graphite being connected by vdW forces. Electrons in the Bi_2_Se_3_ surface specifically demonstrated a semi-metallic, linear dispersion relation found in graphene.

The vdW-layered structures were combined and incorporated into heterostructures with selected 2D materials to increase the photodetector’s performance. In addition, exceptional TI materials have been researched in terms of quantum computing, thermoelectric effects, superconductivity, spin devices, and topological properties. It is believed that more research is required on the TI optoelectronic parameters to be implemented in the detector’s fabrication.

### 7.2. HOT Infrared Topological Insulator Photodetectors

Figure 22 presents the reported utmost current responsivities for TI single-element IR photodetectors [100,101,102,103,104,105,106,107,108] compared with standard InGaAs and HgCdTe photodiodes operating at 300 K [20,69]. The dashed line shows the theoretically simulated ideal photodiode responsivity, exhibiting *η* = 100% and *g* = 1. The variation in the TI-based photodetector’s current responsivity is visible. As reported [103,105,107], the high current responsivities were found to be a result of the photoelectric gain (photogating effect) being observed in LDS to include 2D hybrid detectors—phototransistors (common design) [7] (e.g., combination of the Bi_2_Te_3_ with graphene [105]).

The current responsivity variation is mostly related to the technology status, and TI photodetector research is still under development, facing many challenges, both in mastering reproducibility and the accompanying effects.

Figure 23 compares the *D** versus wavelength for TI IR photodetectors and the current best performance for HOT photodetectors (four families) operating at 300 K: HgCdTe photodiodes, T2SLs III–V IB QCIPs, CQD, and 2D material-based devices. The trend characteristics are presented based on the measured data formerly gathered in Figure 15 and Figure 16 and taken from Refs. [20,110,111,112] for CQD photodetectors. In addition, the standard 300 K thermal detector performance (pyroelectric detector, thermopile, and Golay cell, thermistor bolometer) is also presented.

Compared to the experimental data for the TI IR photodetectors presented in Figure 22, the *D** shown in Figure 23 was found to be less scattered and close to the HgCdTe photodiodes and IB QCIPs trend lines for wavelengths >2 μm. The *D** for TMD photodetectors is higher, but, in this case, as outlined in Section 6.3.2, the high performance partially results from parameter overestimates.

Given the current stage of technology, the *D** for four main HOT detector families presented in Figure 24 was found to be the sub-BLIP, while the TI IR detector’s performance was also lower than the BLIP limit. MCT photodiode technology is highly developed, and the potential properties of HOT MCT photodiodes (“Law 19”) operating in the LWIR range highlight the prospects of reaching *D*,* limited by the scene radiation in the near future. To date, it is also not exactly clarified why the 2D material detectors’ (mainly TMDs) *D** is higher than for HgCdTe photodiodes, if the fundamental MCT properties are reported to be higher [58]. Taking that into consideration, LDS photodetectors that include TI detectors cannot compete with HgCdTe photodiodes and *D**~10^13^ Jones for the 3D Sb_2_Te_3_/n-Si heterostructure for *λ* = 2.4 μm (refer to Figure 23), exceeding the BLIP limit, which is fully overestimated. That may be changed with the TI material technology development and precise monitoring of the material parameters/properties. TIs exhibit several advantages (in terms of the photodetectors’ performance) such as excellent optical properties, exceptional band structures, and high carrier mobility, showing potential for 300 K and wide-spectrum operation, predominantly within the MWIR-FIR regions (Bi_2_Te_3_-Si heterojunction photodetector exhibits photoresponse at 300 K from 370.6 nm/UV to 118 μm/THz [113]); however, two significant limitations may be listed:Issues with the high-quality TI materials’ fabrication, mainly for low-cost synthesis and large scale; so, thorough research and technological progress are indispensable.High dark current stemming from the intrinsic band structure or surface states (volume and topological surface states mixing); a high dark current significantly limits the photodetectors’ performance (possible applications), leading to high power consumption and noise.

The technological issues are mainly connected to the monitoring of the stoichiometric ratio of TI to suppress the defect generation to reduce the dark current and noise. The second main problem is the choice of a suitable compound and the optimal device structure to suppress the dark current contribution and increase the detector’s performance.

To date, the group of TI materials has only been researched theoretically, without experiments proving their capabilities and properties, while connections between the detector’s performance and numerous effects can lead to unique prospects in that subject [109]. To summarize, TIs demonstrate many unique physical properties, enabling photodetection, and further progress is expected in that material family, providing an exceptional approach for next-generation IR HOT detector fabrication. Single-pixel TI detectors’ *D** was found to be close to HgCdTe (dominant material system for IR detector fabrication—60-year tradition) photodiodes for *λ* > 2 μm. At the current level of technology, a high current responsivity variation is reported, which relates to immature device technology and the impact of photogating phenomena (significantly depending on the photodetector’s design, not considered in this paper) observed in the selected LDS photodetectors. What is more, the TI photodetectors have not been fully developed in terms of the optimal device structures and fundamental complex physical effects of clarification to include sophisticated synthesis methods with remarkably elevated levels of accuracy to monitor tunable phase transitions and symmetry manipulations. Finally, the TI detectors’ commercialization is believed to be a distant process, depending not only on the single photodetectors’ performance but also on the large-scale fabrication of high-quality compounds at a low cost. The main aim is to develop TI materials with high-scale integration with electronic/photonic platforms (CMOS technologies).

## 8. Colloidal Quantum Dot (CQD) IR Detectors

The QD photodetectors’ progress (started in the 1990s) evolved from epitaxially grown self-assembled QD photodetectors to a novel family of colloidal nanocrystal-based detectors being observed in the last 10 years, where the absorber is made of 3D nanoparticles [110].

The IR, QDs have been synthesized with selected semiconductor groups to include the following: IV (Ge, Si, GeSn), IV–VI (PbSe, PbS, PbTe), III–V (InSb, InAs), II–VI (HgTe, HgSe), I–VI (Ag_2_Se, Ag_2_S), and I–III–VI (CuInSe_2_, CuInS_2_) [114,115,116]. As of now, research is directed towards Hg- and Pb-based CQDs; however, Hg-based CQDs may theoretically lead to environmental and health issues, leading to a search for alternative materials (e.g., Ag_2_Se CQDs and metal–halide perovskite nanocrystals are considered [109]).

CQDs are considered to be a possible competitor for InSb, InGaAs, InAsSb, MCT, and T2SLs in SWIR and MWIR in terms of the fabrication process (synthesized by liquid-phase chemistry with inexpensive reagents, where many methods have been transferred from the VIS display to include the scalability of the CQD synthesis to limit costs). The most common method to process IR CQDs is a colloidal solution-phase synthesis, where the particle nucleation monitoring in different growth stages in solution containing both metal and anion precursors was found to be the main issue [110].

A review summarizing the recent progress in IR QD development, focusing on fundamental developments in chemistry, synthesis, and characterization methods, was published by Lu et al. [114]. The present research is focused mostly on CQD photovoltaic detectors with suppressed dark currents and 1/*f* noise in comparison to photoconductive devices, which is why that review focused on more complex photodiodes.

The current responsivity of the CQD photodiodes was measured at the level of ~100 mA/W, corresponding to EQE~10–80% being higher in SWIR (see Figure 24). In addition, those EQEs are substantially higher than for epitaxial (self-organized) QD photodetectors, normally reported at the level of ~2%. The standard QEs for market, InSb, InGaAs, HgCdTe, and T2SLs photodetectors, are shown in Figure 24.

**Figure 24 materials-17-04522-f024:**
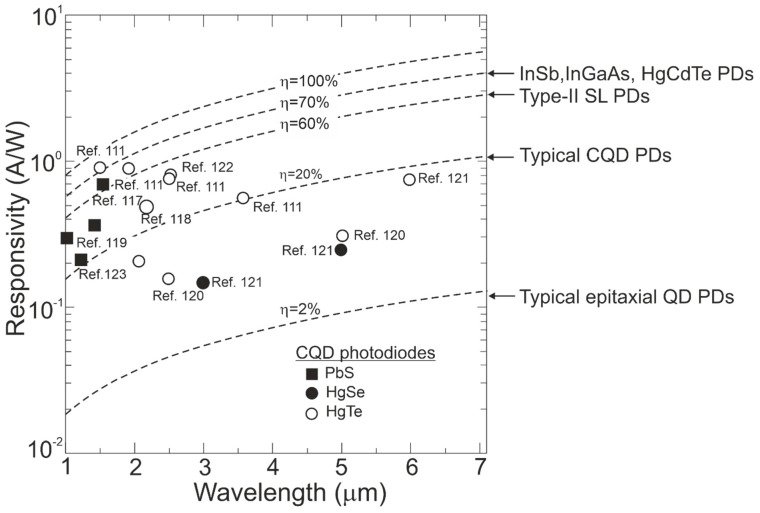
The spectral current responsivity for selected IR technologies at room temperature. The experimental data after Refs. [111,117,118,119,120,121,122,123]. PDs—photodiodes.

The amorphous CQD layers make incorporation with ROIC more simple (monolithic integration into ROIC does not require any hybridization, usually limited to 1-megapixel size), as shown in Figure 25, with no pixel array size restrictions (single pixels are defined by the ROICs metal pads) and a simple production procedure. To produce colloidal nanocrystals, the ingredients are introduced into a flask, and the needed shape and size are reached by controlling the component concentrations, temperature, and ligand selection. That top-surface detector reaches a 100% fill factor (FF) and is well matched with CMOS electronics.

The best-known photodiode design and bandgap structure are shown in Figure 26a. The design with a QD absorber is made of metal oxides, organics (or 2D materials) monolithically integrated with CMOS ROIC by extra-thin-film transport layers. The carriers are moved via an internal electric field in opposite directions through the transport layers. As presented on the right side of Figure 26a, the detectors are produced in either heterojunction designs (bottom diagram) or the homojunction (top diagram).

An increase in absorption by making the CQD layers thicker was reported to be difficult due to the high probability of the low diffusion length of photoexcited carriers in CQD layers and possible crack generation in the CQD film. A small diffusion length is connected to low carrier mobility, reported within a range of several cm^2^/Vs for PbS QDs and <1 cm^2^/Vs for HgTe QDs. To increase light absorption, the CQD thin films were integrated with resonant optical structures (photonic structures to confine and concentrate incident electromagnetic fields) to include back reflectors, gold gratings, and simplified Helmholtz resonators, increasing the QE. Figure 26b shows the HgTe CQD FPA pixel cross-section with the ~400 nm thick absorber and two electrodes—the pixel electrode, acting as electrical contact and back reflector, and the ground electrode (the detector is biased by the pixel and ground electrodes) [112].

The post-fabrication ligand (molecules bound on the CQD surface) exchange mechanism was found to be the most significant stage driving the detector’s ultimate performance in CQD photodetector synthesis, which is related to the high surface-to-volume ratio. It was found that the surface strongly influences the CQD physical parameters, and surface ligands determine CQD’s stability and electrical properties. The main purpose of the ligand exchange mechanism is to transfer the QDs to appropriate ligands and solvents, control the majority carrier type, and adjust/increase the carrier mobility. It was proved that efficient carrier transport can be achieved by replacing of the isolating long organic ligands by small and/or ionic ligands. Effective ligand exchange paths are reported in Refs. [25,117].

The PbS CQDs operating in 1–3 μm NIR represent the lead chalcogenide group [110,124], with *D** reaching the InGaAs photodiode level (refer to Figure 2 and Figure 15). The 100 nm thick PbS CQDs were reported to be incorporated as effective absorbers [125].

HgTe CQDs were reported to be operating within SWIR-THz [112]. Figure 27a presents the HgTe CQD PV detector on an ITO/sapphire substrate, where the ~400 nm thick absorber was deposited layer by layer using the drop-casting method. The ~10 nm Ag_2_Te nanoparticles were implemented to create a p-doped HgTe CQD top layer. Figure 27b presents *D** versus temperature for selected material systems (*λ_c_*~5 μm) to include well-developed HgCdTe and HgTe CQD photodiodes with proper experimental data. *D** was improved (2/3×) for the device supported by a plasmonic nano-disc compared to the detector without an absorption boost. The assessed *D** for CQD is lower than HgCdTe photodiodes. For *T* > 200 K, the theoretically calculated *D** for MCT photodiodes is restricted by the scene (refer to the considerations given in Section 6.2.3—“Law 19”). Lately, the HgTe-based CQD gradient homojunction photodiodes [refer to Figure 27b], synthetized by upgrading the ligand exchange method, reached even better rectification behavior through gradual doping-level changes, leading to an increased *D** > 10^9^ Jones for *λ_c_*~4 μm and 300 K [25].

Figure 28 shows the highest reported *D** for selected detectors (PC-PbSe, PC-PbS, PV-InGaAs, PV-Si, PV-MCT, and CQD IR photodiodes) operating at 300 K. The results indicate the sub-BLIP conditions and the performance tendency curves for MCT photodiodes. IB QCIPs and 2D material detectors were also shown for comparison reasons based on the measured data gathered in Figure 15. The CQD detectors’ *D** in the MWIR region is comparable to MCT photodiodes, suggesting significant progress in CQD development that is remarkably visible in SWIR, where the CQD photodetectors’ performance is even higher. The CQD photodiodes’ high *D** in SWIR and MWIR does not correspond to the high dark current densities depicted in Figure 13. As shown in this figure, the CQD detector current densities are higher than those reported for MCT photodiodes.

The majority of CQD detector research has been focused on single-pixel devices, but CQDs also offer potential as a possible replacement in the typical image sensor field, advancing from solution processability, Si compatibility, and simplified device fabrication [112,125,129]. Many studies about SWIR and MWIR imagers have been reported, with a possible extension of the spectral range to LWIR.

The very first high-quality CQD PbS array was presented by RTI International, based on a heterojunction: C60 (n-type) active layers and PbS-CQDs (p-type) [30,130]. The first SWIR cameras made of CQD thin-film photodiodes processed monolithically on Si ROIC have been reported [129,131]. Table 8 reviews the performance of CQD sensors. The monolithic processing provides affordability for many applications, such as the medical, smart agriculture, surveillance, and automotive sectors.

Emberion demonstrated its broad spectral 400–2000 nm VIS-SWIR camera Emberion VS20 in 2021. Acuros^®^ SWIR cameras (Durham, NC, USA) developed by SWIR Vision Systems are designed for high-resolution images and provide a high dynamic range, high frame rates, and a broad range of exposure times. The 1920 × 1080 (2.1 megapixels, 15 μm pixel pitch) Acuros cameras are fabricated at a higher cost in comparison with InGaAs SWIR cameras (Stadtroda, Germany). The IMEC’s imager has a resolution of 758 × 512 pixels and 5 µm pixel pitch. The CQD photodiodes on the Si substrate reach EQE > 60% for 940 nm and ~40% for 1450 nm, allowing for uncooled operation with the dark current, comparable to the market InGaAs detectors. ST Microelectronics demonstrated global snapshot PbS CQD image sensors fabricated on a 300 mm Si wafer, with the smallest NIR pixel (1.62 μm) and highest QE~60%. The technology is expected to be able to provide high-quality full spectral responsivity (from UV to SWIR) in mobile devices, vision systems, mini-spectrometers/hyperspectral imaging, and automotive advanced driver assistance systems (ADASs).

Generally, CQD SWIR cameras offer a wide dynamic range, owing both to the low noise and non-saturating characteristics of the detector. Figure 29 compares images obtained from the CMOS InGaAs camera (left) and SWIR Emberion camera (right). The contrast differences in the clouds, balconies, and car windows are observed by the Emberion CQD sensor.

CQD PbS detectors cover IR < 3 μm, with low *D** for < 2 μm. Buurma et al. [134] reported the HgTe CQD-based array covering MWIR. The monolithic array, with 320 × 256 and a pixel pitch of 30 μm, was fabricated with FLIR ISC9809 ROIC, reaching a median *NEDT* of 102 mK at a 100 K operating temperature (poor-quality image). Recently, a research group from the Beijing Institute of Technology demonstrated significant advances in HgTe CDQ photodiodes operating at ~2 μm and 300 K (see Table 8).

Currently, CQD cameras are being implemented in applications requiring high-definition low-cost imaging based on smaller pixels without strong sensitivity. It is predicted that increasing the dot size and maintaining a proper mono-dispersion, carrier transport, and QE will improve the noise levels. Finally, the continuous development of deposition and synthesis techniques will allow us to reach much higher device performance in the future.

## 9. Conclusions

Over the past three decades, we have seen the development of a new class of photodetectors in a wide spectral range, from the UV to the FIR, whose active sensing regions are LDS. These include, in particular, 2D materials, perovskites, and QDs/NWs. The performance of those devices competes with standard commercial photodetectors available on the global market. Their performance beyond physical limitations is reported/cited.

In the UV range, apart from such wide-bandgap semiconductors as SiC, AlGaN, and diamond, Ga_2_O_3_ has become a key material due to the very high breakdown electric field and the availability of large-diameter wafers. Ga_2_O_3_ photodetectors are fabricated from materials in a variety of forms, from single crystals and epitaxial layers to nanostructures. However, the detector parameters (responsivity, response speed, detectivity) are scattered over a wide range, which is related to immature technology, different architectures, different material phases, and crystal properties. Moreover, over the past few years, great progress in improving the Ga_2_O_3_ photodetectors’ performance has been observed, with record detectivities, even exceeding the physical limit of detection (SFL).

Perovskites have attracted wide research interest due to direct bandgaps, long carrier diffusion lengths, high absorption coefficients, and high defect tolerance. This development was mainly triggered by the revolutionary evolution of the solid-state perovskite solar cell as a strong candidate for a next-generation solar energy harvester and other device applications, including photodetectors, mainly in the VIS and NIR bands. The best figure of merit of perovskite photodetectors indicates the strong competition for Si-based devices in sensing and imaging applications.

From an application point of view, perovskites are generally not as stable as inorganics due to their ionic nature. In a humid environment, a drastic change in the crystalline structure and even the composition, causing irreversible damage to the material, was reported. However, recently published results suggest that perovskite devices are “not genetically defective” in terms of stability [135]. Toxicity is another issue. So far, lead-based perovskites provide devices exhibiting the highest performance, and the further expansion of perovskite materials in optoelectronic applications is still ahead of us.

In the IR region, MCT is the most widely used variable-gap semiconductor and is considered a reference for alternative technologies. For more than six decades, MCT has efficiently coped with major challenges from detector families, including extrinsic silicon, lead–tin telluride devices, Schottky barriers on silicon, AlGaAs multiple quantum wells, T2SL, and, especially, silicon microbolometers. In the last decade, LDSs (mainly 2D materials) have been found to be an alternative to MCT. Currently, MCT has more competitors than ever before [15]; thus, a timely question needs to be answered: *will the discovery of LDS materials affect the status of MCT?* Based on the results presented in this study, the following conclusions can be drawn:In spite of sixty years of HgCdTe development, its ultimate BLIP performance limit at 300 K has not been reached. It requires a doping concentration of about 10^13^ cm^−3^ in a photodiode absorber, together with a millisecond carrier lifetime. That doping concentration level has been recently reached in totally depleted P-i-N photodiodes by Teledyne and DRS.The potential properties of HOT MCT photodiodes (“Law 19”) operating in a wavelength range of >4 μm guarantee reaching the BLIP condition, which is the main goal in IR technology.The T2SL IB QCIP performance is comparable with MCT photodiodes and proved to be operating at >300 K; however, challenging technology and high fabrication costs prevent their progress.The extraordinary and unique electronic and optical 2D material properties could lead to a promising alternative for IR detectors. Although remarkable TMD performance (utmost detectivity) has been reported at 300 K and in the LWIR region, many challenges must be faced to fully exploit the distinct advantages of those materials.The 2D material-based detector performance was reported to be often overestimated due to miscalculation of the device active area and light power density, wrong noise estimates and contradictory bandwidth assumed for measured noise, and responsivity.Two-dimensional materials exhibit short carrier lifetimes and high optical absorption. Several methods, including the photogate effect with a graphene fast transfer channel and electron trap layers, can be implemented to improve the sensitivity, but the carrier mobility and response time limit practical applications (tradeoff between sensitivity and response time).TI single-pixel detectors detectivities are close to the MCT photodiode tendency curves for >2 μm.The CQD performance is lower than technologies considered in this study, except for PbS CQD-based devices, exhibiting detectivity comparable with InGaAs.CQD-based FPA could be fabricated at a low cost.

In summary, MCT will remain a material of choice for IR detector fabrication in the near future. None of the alternative technologies can compete in terms of the fundamental physical properties crucial to fabricate high-performance devices. Currently, “*third wave*” materials may be more flexible in the fabrication process, but they never provide higher performance or, except thermal detectors, operate at higher or comparable temperatures.

To confirm the above predictions, it is worth noting that, lately, completely depleted 640 × 512 P-i-N HgCdTe photodiode FPAs (in MWIR and LWIR operating up to 250 K and 160 K) have been reported [58,59,136]. Further technological improvements in P-i-N MCT-depleted photodiodes are believed to increase the operating temperature to 300 K in both MWIR and LWIR devices [74].

## Figures and Tables

**Figure 1 materials-17-04522-f001:**
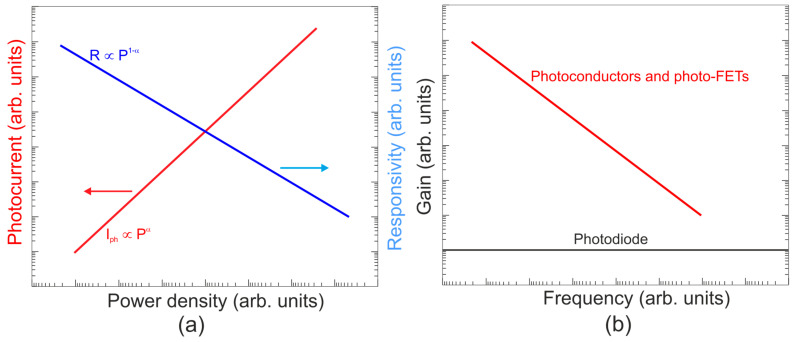
The net photocurrent and responsivity radiation power dependence (**a**), and gain versus frequency (**b**) for photodetectors.

**Figure 2 materials-17-04522-f002:**
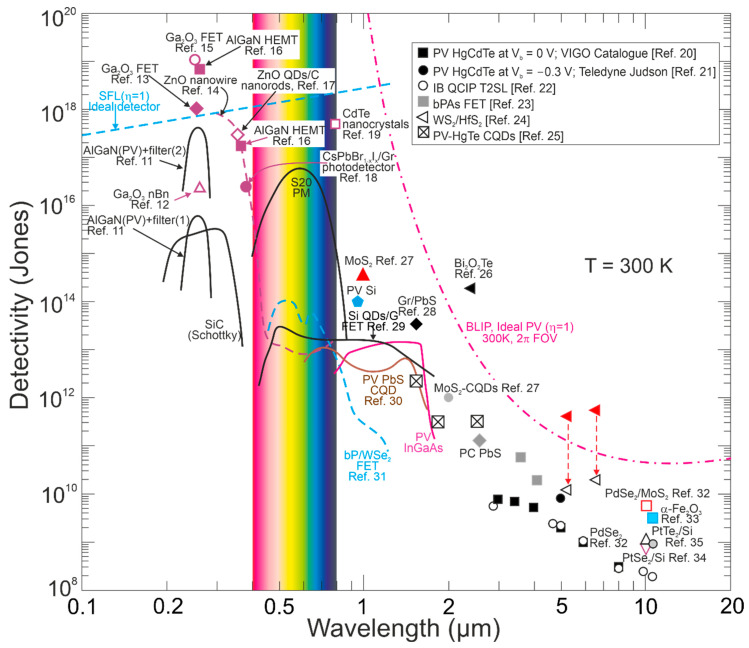
The highest detectivity reported for photodetectors operating at 300 K within the wavelength range 0.1–20 μm compared with the ultimate SFL and BLIP. PC—photoconductive detector; PV—photovoltaic detector; FET—field-effect transistor; and PM—photomultiplier. The measured data for typical photodiodes and LDS photodetectors are taken after the Refs. [11,12,13,14,15,16,17,18,19,20,21,22,23,24,25,26,27,28,29,30,31,32,33,34,35].

**Figure 3 materials-17-04522-f003:**
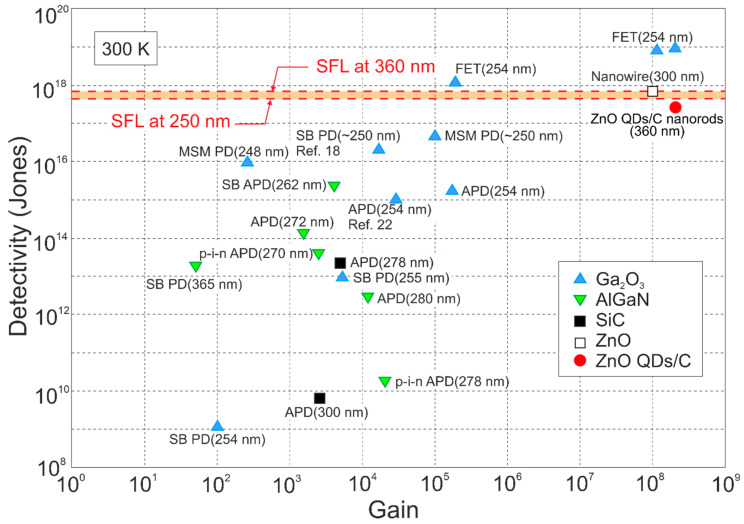
Detectivity versus gain for selected UV photodetectors at 300 K (after Refs. [4,18,22]). Theoretical predictions for SFL within wavelength range 250–360 were presented for the reference. FET—field-effect transistor, PD—photodiode, SB—Schottky barrier, MSM—metal–semiconductor–metal, APD—avalanche photodiode, QDs—quantum dots.

**Figure 4 materials-17-04522-f004:**
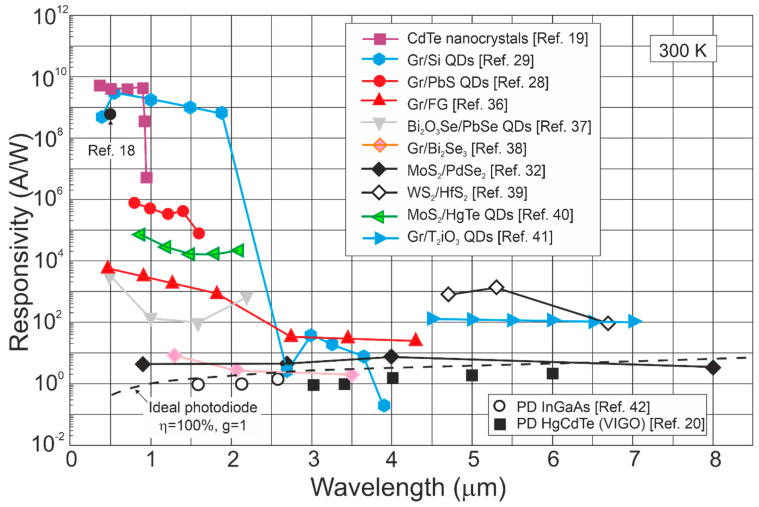
Spectral responsivity for LDS IR photodetectors exhibiting high photoelectric gain at 300 K (according to the Refs. [18,19,20,28,29,32,36,37,38,39,40,41]). For comparison reasons, the responsivity for InGaAs (according to Ref. [42]), HgCdTe photodiodes and theoretical responsivity curves is shown.

**Figure 5 materials-17-04522-f005:**
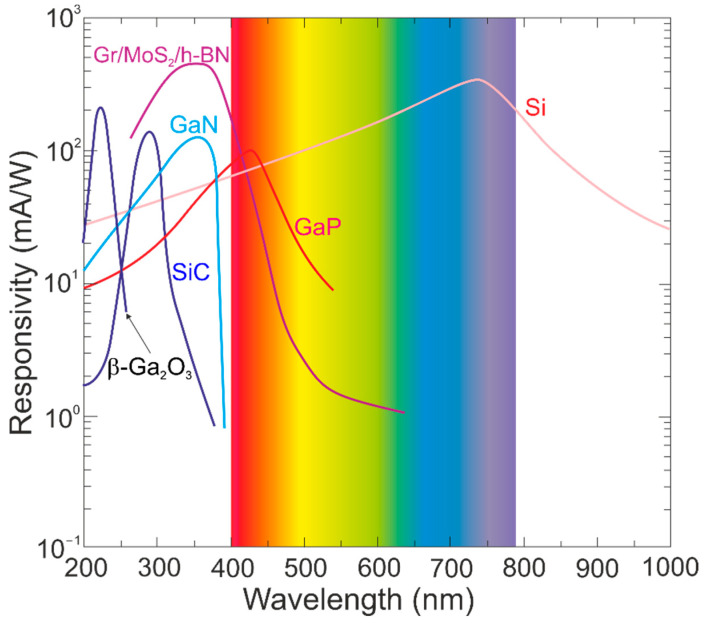
Spectral responsivity for the PV β-Ga_2_O_3_, hybrid Gr/MoS_2_/h-BN and commercial photodiodes [GaP-PD (PDA25K2, Thorlabs), Si-PD (PDA10A2, Thorlabs), GaN-PD (G365S01S, Hefei Photosensitive Semiconductor), SiC-PD (18ISO90, Boston Electronics)] UV photodetectors [44].

**Figure 6 materials-17-04522-f006:**
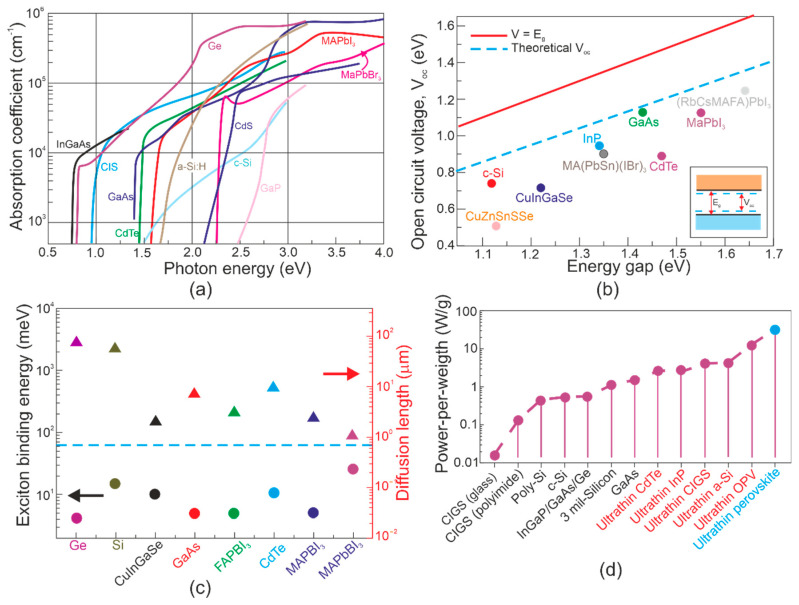
The main material parameters for thin-film solar cell’s fabrication: (**a**) absorption coefficient, (**b**) open-circuit voltage (*V_oc_*) for commercially available technologies, (**c**) diffusion length/binding energy, (**d**) power per weight for principal lightweight solar cells (after Ref. [46]).

**Figure 7 materials-17-04522-f007:**
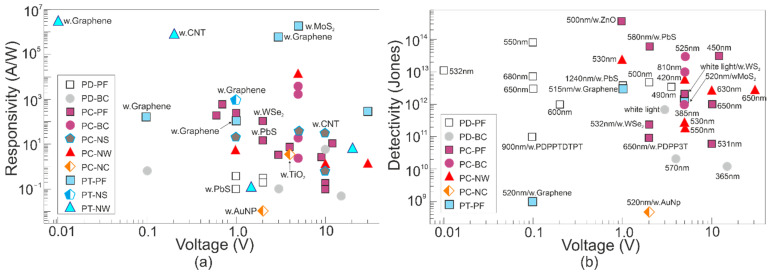
Current responsivity (**a**) and detectivity (**b**) versus voltage for selected perovskite material-based devices. PC—photoconductor; PT—phototransistor; PD—photodiode; BC—bulk crystal; PF—polycrystalline film; NW—nanowire; NC—nanocrystal; NS—nanosheet (after Ref. [49]).

**Figure 8 materials-17-04522-f008:**
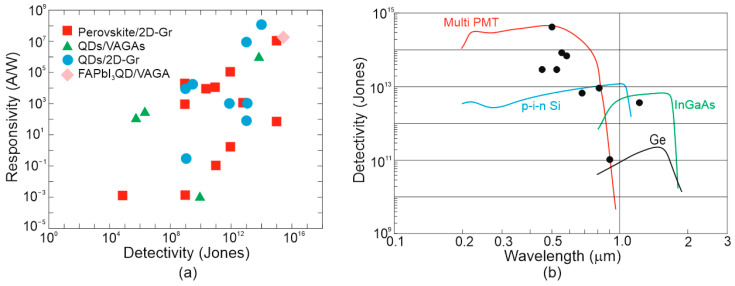
The 300 K performance of selected perovskite photodetectors: (**a**) detectivity dependence on responsivity for selected hybrid photodetectors (after Ref. [50]); (**b**) spectral detectivity comparison [data presented in Figure 7b] with commercially available Si, Ge, InGaAs photodiodes and Multi PMT. PV—photovoltaic detector; PM—photomultiplier; FET—field-effect transistor; QDs—quantum dots.

**Figure 9 materials-17-04522-f009:**
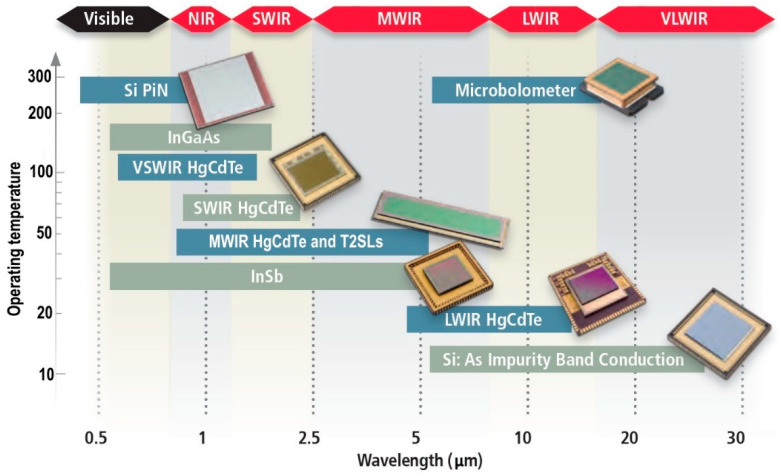
The most common materials for IR detectors’ fabrication in terms of operating temperature and wavelength.

**Figure 11 materials-17-04522-f011:**
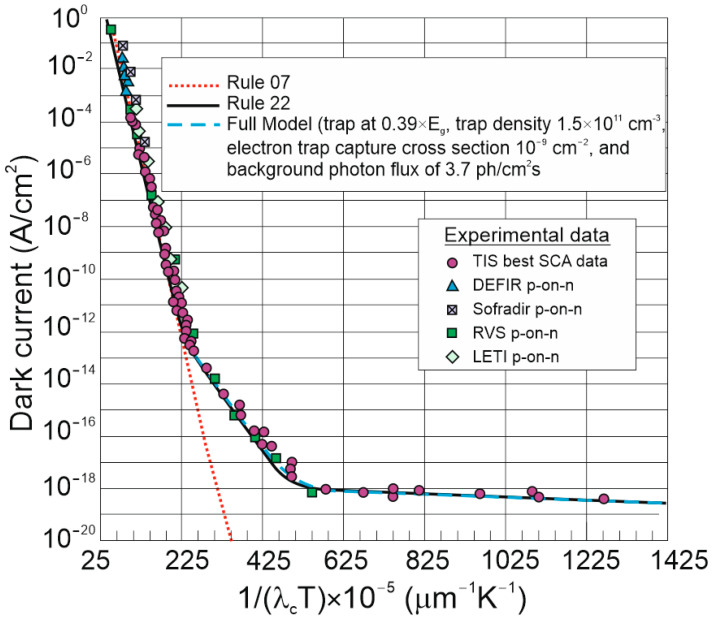
Dark current for the highest-performance p-on-n FPAs developed by Teledyne Imaging Sensors (TIS) and ultimate performance reported for HgCdTe detectors (after Ref. [68]).

**Figure 12 materials-17-04522-f012:**
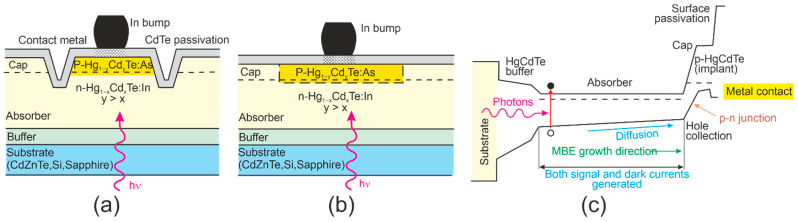
The HgCdTe DLHJ developed by Teledyne: (**a**) mesa design, (**b**) planar design, and (**c**) bandgap structure.

**Figure 13 materials-17-04522-f013:**
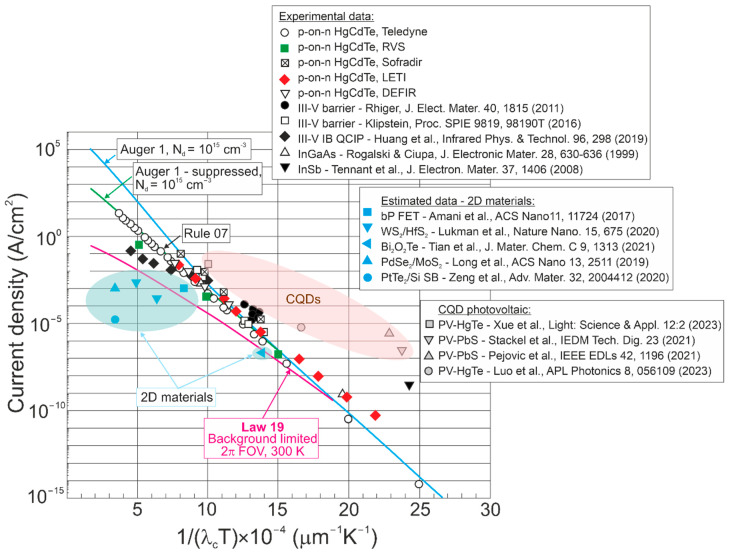
The HgCdTe p-on-n photodiodes current density versus 1/(*λ_c_T*) (based on Ref. [59]). The experimental data are collected for p-on-n MCT photodiodes [61,68] and selected technologies to include 2D materials and CQDs.

**Figure 14 materials-17-04522-f014:**
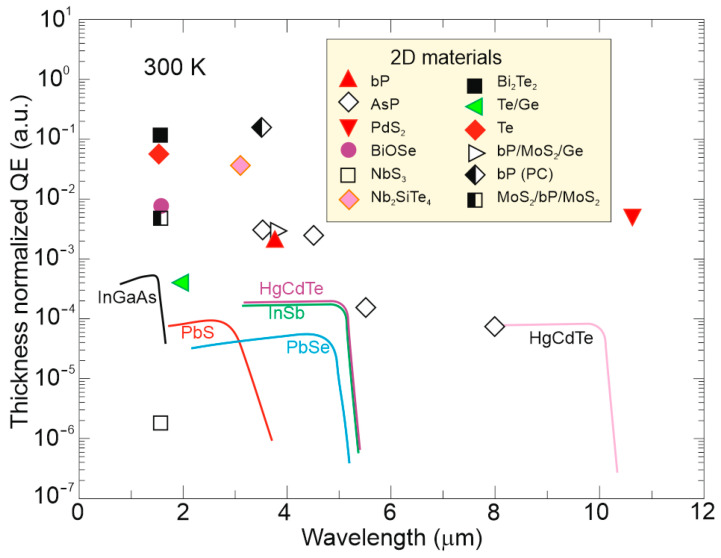
Thickness-normalized external quantum efficiency for selected 2D materials and typical thin-film materials (after Ref. [72]).

**Figure 15 materials-17-04522-f015:**
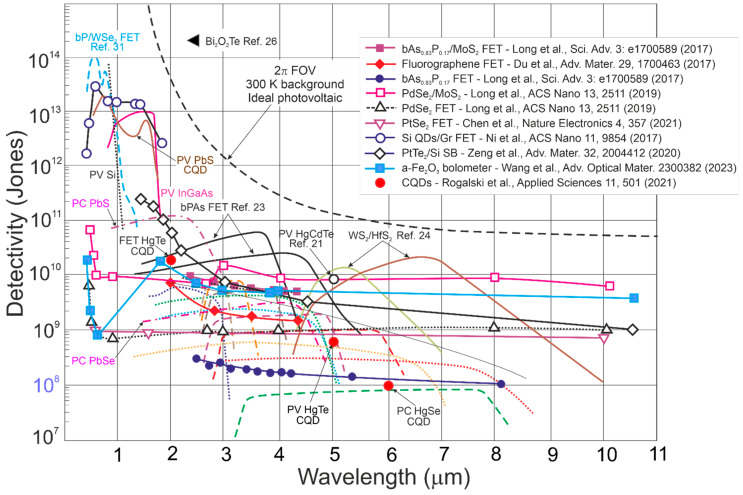
Detectivity versus wavelength for selected 2D material-based detectors (Refs. [23,24,31]) and commercially available photodetectors for 300 K [PV-InGaAs and PV-Si, PC-PbSe and PC-PbS, MCT photodiodes (dashed lines—Ref. [20], Ref. [21]). T2SL IB QCIPs are marked by dotted lines (Ref. [39]). PC—photoconductor, PV—photodiode, FET—field-effect transistor.

**Figure 17 materials-17-04522-f017:**
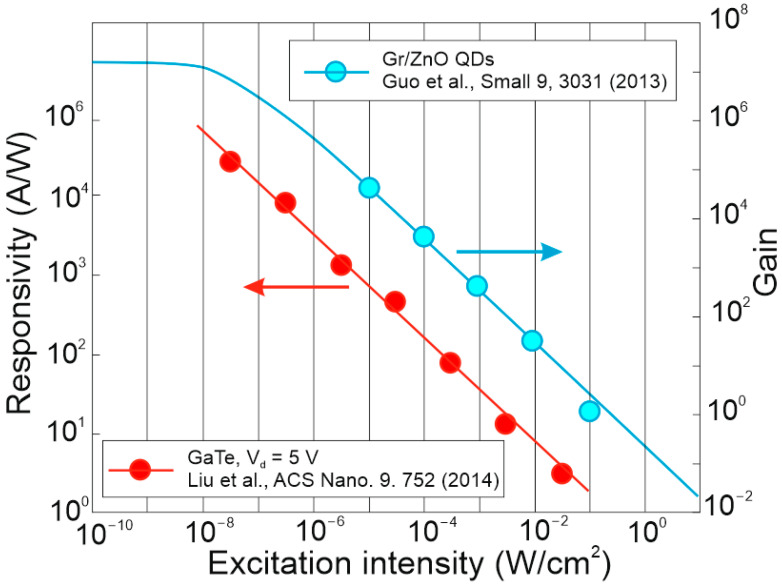
Responsivity and gain versus excitation intensity for multilayer GaTe flakes [79] and hybrid graphene/ZnO QDs detectors [80]. The red and blue circles represent the experimental data, and the solid curves show theoretical, best-fitting plots.

**Figure 18 materials-17-04522-f018:**
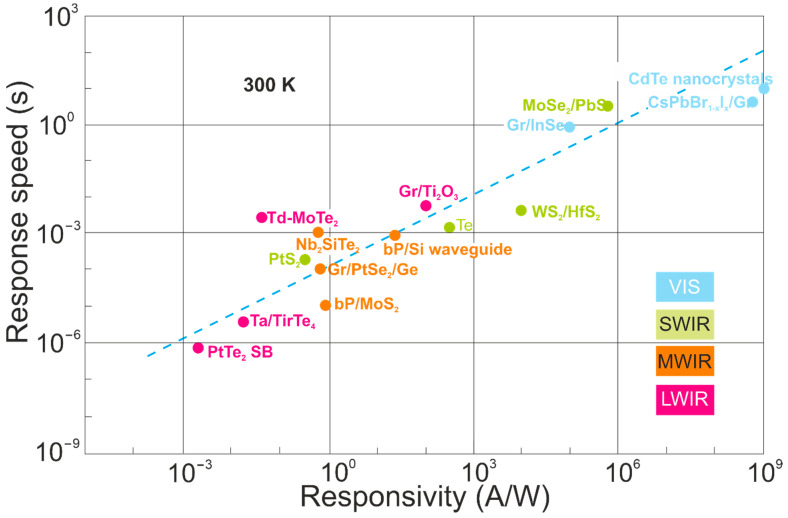
The current responsivity dependence on the response time for 2D material-based detectors operating within VIS, SWIR, MWIR and LWIR at 300 K. Experimental data were taken from selected papers (after Ref. [81]).

**Figure 19 materials-17-04522-f019:**
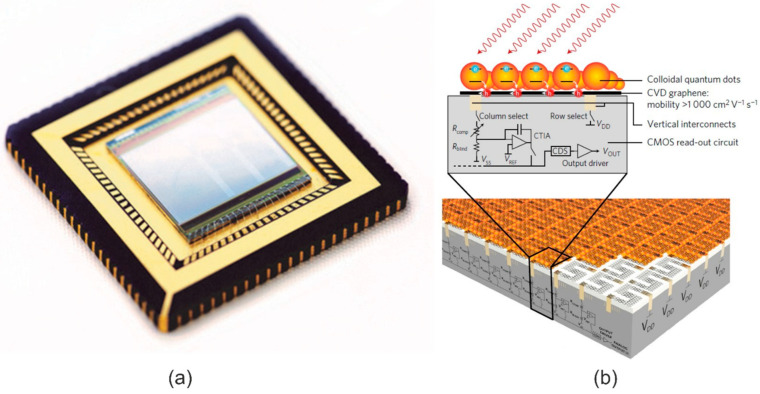
Monolithically integrated graphene/CQDs array (**a**), detector’s side view and ROIC (**b**) (after Ref. [84]).

**Figure 20 materials-17-04522-f020:**
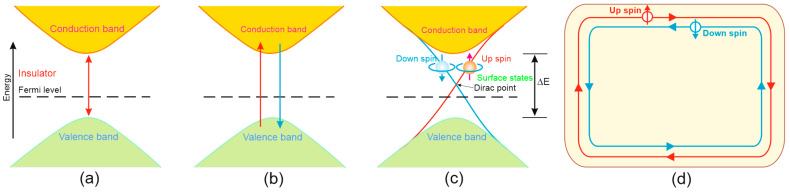
TI: (**a**) Large bandgap of typical insulator, (**b**) influence of strong spin–orbit coupling, (**c**) 3D TI theoretical band structure, and (**d**) theoretical spin-resolved band structure of the edge states.

**Figure 21 materials-17-04522-f021:**
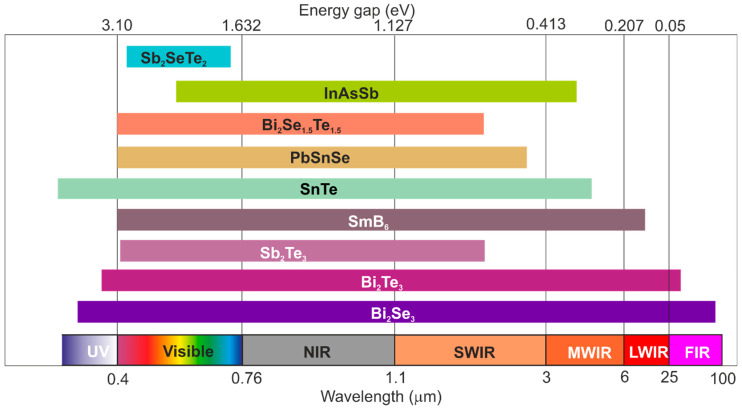
Selected TI materials versus operating wavelength.

**Figure 22 materials-17-04522-f022:**
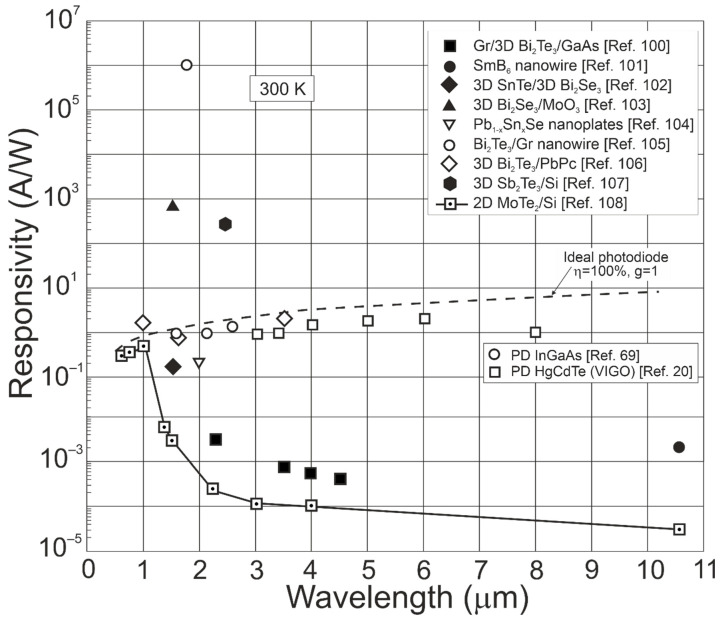
The current responsivities for selected TI IR photodetectors (Refs. [100,101,102,103,104,105,106,107,108]) and InGaAs and HgCdTe photodiodes at 300 K (Refs. [20,69]). The dashed curve depicts spectral responsivity for perfect photodiode exhibiting *η* = 100% and *g* = 1 (after Ref. [109]).

**Figure 23 materials-17-04522-f023:**
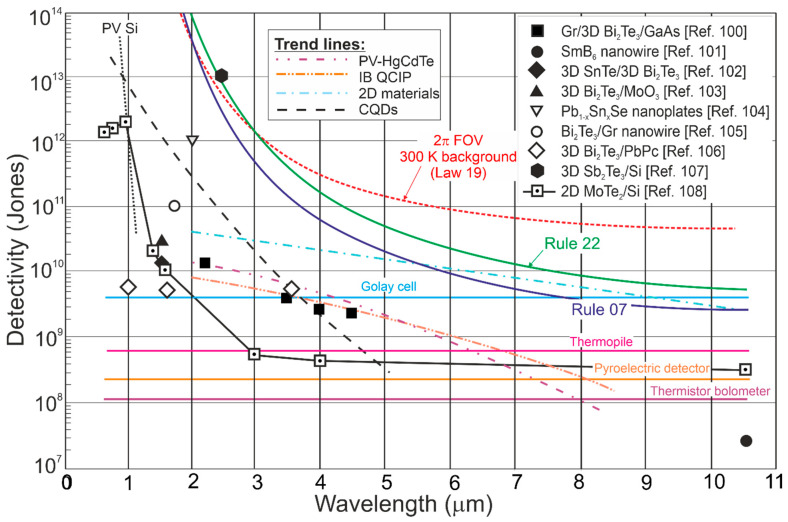
*D** versus wavelength for TI IR photodetectors (Refs. [100,101,102,103,104,105,106,107,108]) compared with selected HOT photodetectors operating at 300 K: HgCdTe photodiodes, T2SLs III–V IB CIPs, 2D material and CQD photodetectors. Simulated curves for utmost BLIP conditions (2π FOV, 300 K scene) and specified by both “Rule 07”/“Rule 22” are depicted (after Ref. [109]).

**Figure 25 materials-17-04522-f025:**
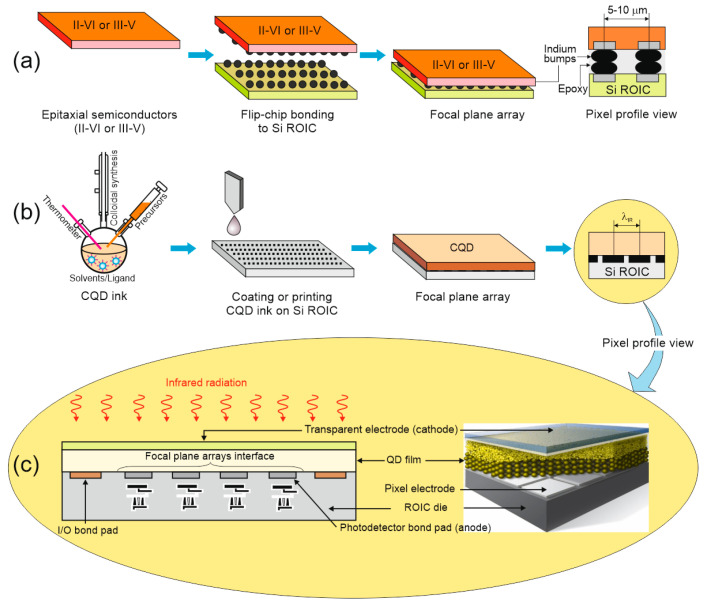
A comparison of the hybrid (smallest pixel pitch of hybrid arrays ~5 μm) integrated III–V and II–VI semiconductor arrays processing (**a**) and CQD monolithic arrays (**b**). Cross-section of CQD monolithic array structure with potential pixel pitch < 1 μm (**c**).

**Figure 26 materials-17-04522-f026:**
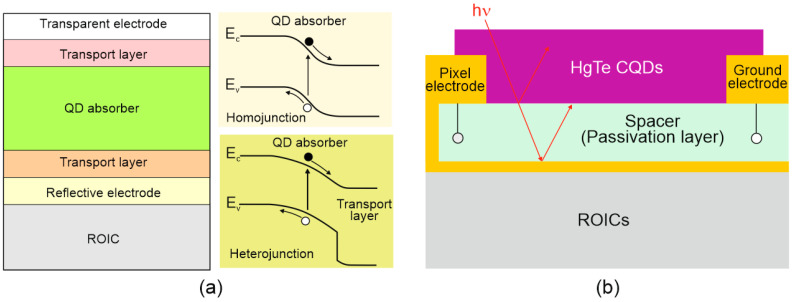
The CQD-based detector structure: (**a**) device design with bandgap energy for homojunction and heterojunction, (**b**) cross-section of HgTe CQD resonant-cavity-enhanced pixel.

**Figure 27 materials-17-04522-f027:**
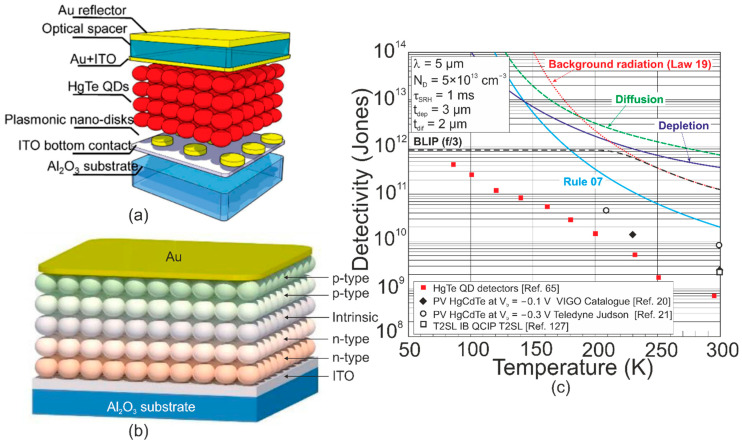
The MWIR HOT HgTe CQD-based detector: (**a**) design with supported plasmonic discs (after Ref. [126]), (**b**) homojunction P-i-N photodiode cross-section (after Ref. [25]), and (**c**) theoretically (“Diffusion” and “Depletion”) simulated *D** versus *T* for MWIR P-i-N photodiode for *τ_SRH_* = 1 ms active layer doping 5 × 10^13^ cm^−3^. Absorber thickness, *t* = 5 μm corresponding to: *t_dif_* = 2 μm and *t_dep_* = 3 μm. The experimental data extracted from Refs. [20,21,65,127]. PV—photodiode, PC—photoconductor, T2SL (type-II superlattice) IB QCIP—interband cascade infrared photodetector (after Ref. [66]).

**Figure 28 materials-17-04522-f028:**
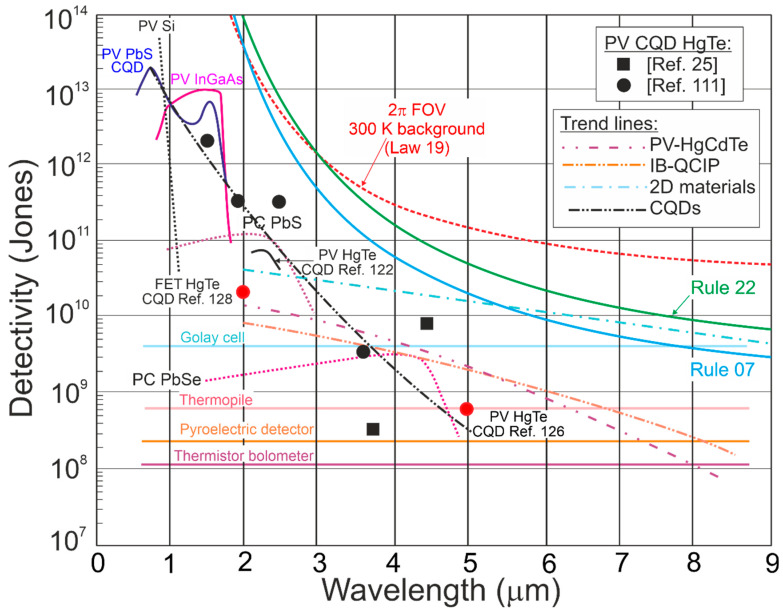
Room-temperature detectivity for CQD photodiodes [25,111,122,126,128] and market photodetectors [PC-PbS and PbSe, PV-Si and InGaAs, MCT photodiodes (solid lines]. The tendency curves for IB QCIPs, MCT photodiodes, 2D material and CQD detectors are also shown for comparison reason based on the data depicted in Figure 15. PV—photodiode, PC—photoconductor.

**Figure 29 materials-17-04522-f029:**
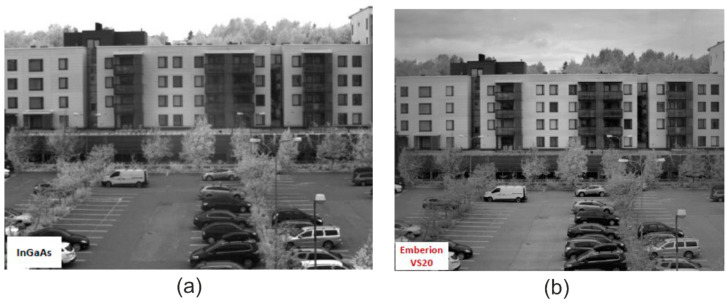
The image comparison taken by SWIR CMOS InGaAs camera (**a**) and Emberion camera (**b**) (after Ref. [132]).

**Table 1 materials-17-04522-t001:** Photon detectors.

Detector	Design	Energy Band Diagram
**Photoconductor** Radiation-sensitive photoresistor. Electron-hole pairs generated in a homogeneous semiconductor. Electric field applied across the absorbing region causes a current to flow being proportional to the incident light (assuming the photon energy exceeds bandgap energy).	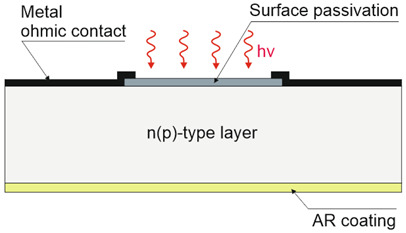	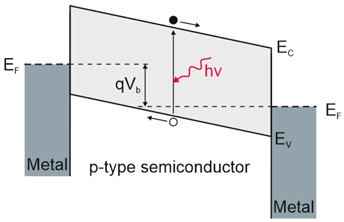
**P-N photodiode** p-on-n/n-on-p the common design. Electric field between n and p regions (depletion region) separates photogenerated carriers on proper part of the p-n junction. Potential barriers drive oppositely charged carriers into opposite directions depending on the external polarization. The photogenerated current influences the short-circuit junction current and the open-circuit junction bias.	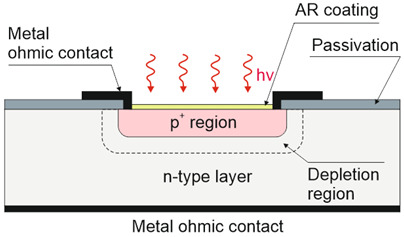	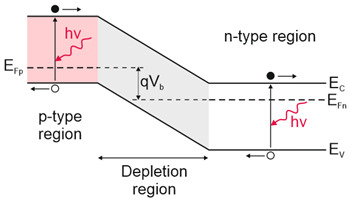
**P-i-N photodiode** P-i-N diode consists of an intrinsic i-region inserted between the N and P layers (capital letter-wide bandgap). The depletion layer extends over entire intrinsic region for the reverse bias. Photons generate the electron-hole pairs only in this region. Without electrically neutral region, the zero diffusion current in the detector is visible, and minority carriers generated via defect centers in the depletion layer contribute to the dark current.	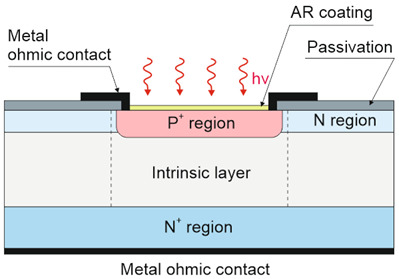	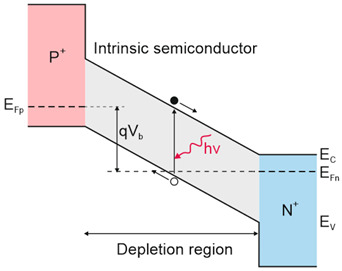
**Avalanche photodiode (APD)** When a reverse voltage is applied to the device, the generated carriers are accelerated by the electric field gaining enough energy to create extra electron-hole pairs via impact ionization and leading to a significant increase in the output signal, improving the APD’s net sensitivity. The APD exhibits fast response time, high quantum efficiency and sensivity.	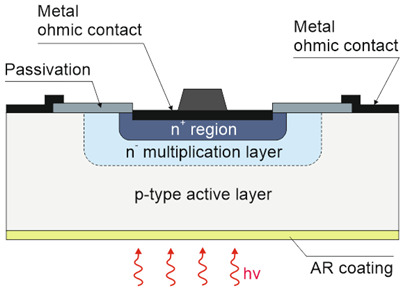	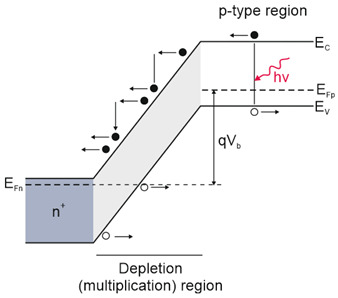
**Schottky barrier photodiode** The Schottky barrier photodiode is built of a metal-semiconductor junction to separate and collect the photogenerated charge carriers. The carriers generated in the depletion layer are effectively removed by the built-in electric field resulting in the photocurrent. Schottky barrier photodiode exhibits advantages (compared to a p-n) to include simple processing/fabrication and a higher reaction rate, however, exhibits significant dark current.	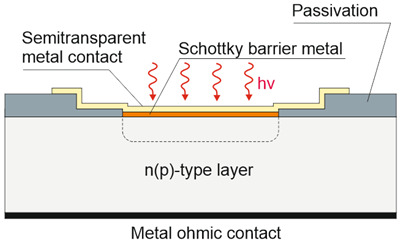	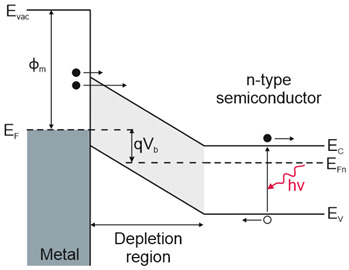
**Metal–semiconductor–metal (MSM) photodiode** Similar to the interdigitated photoconductor—where ohmic contacts are replaced by the Schottky barriers building the metal-semiconductor and semiconductor-metal junctions allowing to reach lower dark current comparing to Schottky photodiode and exhibits a faster response time than a P-i-N.	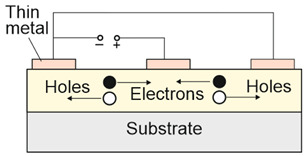	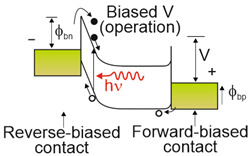
**FET photodetector** The detector structure resembles lateral photoconductor—MSM forming the drain and source electrodes. The modulation of the channel conductivity is provided by an extra gate electrode being separated by a thin dielectric layer. The gate voltage (*V_G_*) controls the carrier density by the field modulation effect and favorably controls dark current by driving the detector into the depletion regime. The light conditions (carriers photogeneration) trigger the device channel conductivity benefiting from the photoelectric gain similarly to the photoconductors.	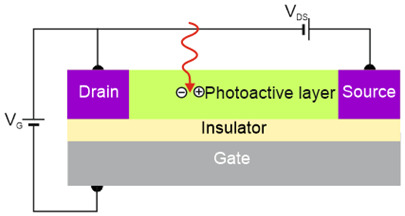	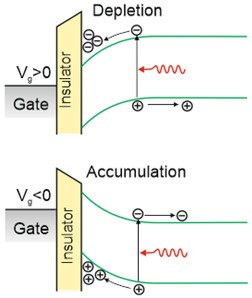

**Table 2 materials-17-04522-t002:** The comparison of photoconductive (PC) and photovoltaic (PV) detectors.

Parameter	PC Detector	PV Detector	Schematic Figures
Gain (*g*)	$g=\frac{\tau }{\tau_{t}}=\frac{\tau \mu_{e}V_{b}}{l^{2}}$ ^(1)^	g=1 (for APD>>1) ^(2)^	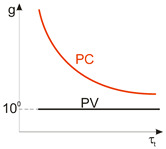
Responsivity (*R*)	$R_{v}=\frac{V_{s}}{P_{\lambda }}=\frac{\eta }{lwt}\frac{\lambda \tau }{hc}\frac{V_{b}}{n_{o}}$ ^(3)^	$R_{i}=\frac{I_{s}}{P_{\lambda }}=\frac{q\eta \lambda }{hc}$ ^(4)^	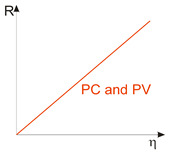
Noise	$\bar{i_{gr}^{2}}=\frac{4qIg∆f}{1+\omega^{2}\tau^{2}}$ ^(5)^	$\bar{i_{sh}^{2}}=2qI∆f=2q\left(I_{0}e^{qV/kT}+I_{0}\right)∆f$ ^(6)^	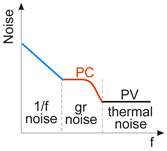
Noise equivalent power (*NEP*)	$NEP=\frac{v_{n}}{R_{v}}$ ^(7)^	$NEP=\frac{i_{n}}{R_{i}}$ ^(8)^	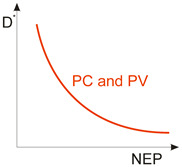
Detectivity (*D**)	$D^{*}=\frac{{\left(A∆f\right)}^{1/2}}{NEP}=\frac{R_{v}{\left(lw∆f\right)}^{1/2}}{{\left(\bar{v_{J}^{2}}+\bar{v_{gr}^{2}}\right)}^{1/2}}$ ^(9)^	$D^{*}=\frac{{\left(A∆f\right)}^{1/2}}{NEP}=\frac{\eta \lambda q}{hc}{\left(\frac{4kT}{R_{0}A}+2q^{2}\eta \phi_{b}\right)}^{-1/2}$ ^(10)^	
BLIP detectivity	$D_{BLIP}^{*}=\frac{\lambda }{2hc}{\left(\frac{\eta }{\phi_{B}}\right)}^{1/2}$	$D_{BLIP}^{*}=\frac{\lambda }{hc}{\left(\frac{\eta }{{2\phi }_{B}}\right)}^{1/2}$	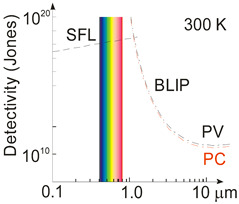
SFL detectivity	$D_{SFL}^{*}=\frac{\eta \lambda }{2^{3/2}hc}{\left(\frac{A}{\Delta f}\right)}^{1/2}$	$D_{SFL}^{*}=\frac{\eta \lambda }{2^{3/2}hc}{\left(\frac{A}{\Delta f}\right)}^{1/2}$	

^(1)^ *τ*—carrier lifetime, *τ_t_* —carrier transit time, *l*—carrier transit length, *m_e_*—carrier mobility, *V_b_*—voltage; ^(2)^ APD—avalanche photodetector; ^(3)^
*V_s_*—the output rms voltage, *P_λ_*—incident irradiance power, *η*—quantum efficiency (QE), *w*—detector width, *t*—detector thickness, *λ*—incident light wavelength, *h*—Planck constant, *c*—light speed, *n*_0_—majority carrier concentration in the n-type material; ^(4)^
*I_s_*—output rms current, ^(5)^
*q*—electron charge, $\bar{i}$—average current, *g*—PC gain; Δ*f*—bandwidth; ^(6)^
$\bar{i_{sh}}$—shot noise, *I*—total current; *V*—voltage; *k*—Boltzmann constant, *T*—operating temperature; ^(7)^
*v_n_*—rms noise voltage, *R_v_*—voltage responsivity; ^(8)^
*i_n_*—rms noise current, *R_i_*—current responsivity; ^(9)^
*A*—detector’s photosensitive area, $\bar{v_{J}}$—Johnson noise voltage, $\bar{v_{gr}}$—generation-recombination noise voltage; ^(10)^ *R*_0_*A*—product of zero-bias resistance and photosensitive area; *ϕ_b_*—the background radiation flux density.

**Table 3 materials-17-04522-t003:** The selected materials used for UV photodetectors’ fabrication.

Parameters	Diamond	Si	4H-SiC	6H-SiC	AlN	MgO	β-Ga_2_O_3_	GaN	ZnO	ZnSe	MgO
Bandgap energy (eV)	5.45	1.12	3.26	3.02	6.2	7.83	4.85	3.4	3.37	2.58	7.83
Density (g/cm^3^)	3.52	2.329	3.21	3.21	3.32	3.58	5.95	6.15	5.61	5.42	3.58
Thermal conductivity (W/cmK)	22.9	1.45	3.7	4.9	4.6	4.82	0.20	3.2	5.4	0.18	4.82
Melting point (K)	3773	1683	3100	3100	>2500	3073	1795	>2000	2242	1517	3073
Electron saturation velocity (10^5^ m/s)	2.3	1	2.1	2	1.3		1.1	1.4			
Electron mobility (cm^2^/Vs)	7300	1240	950	400	420	10	300	1000	170	540	10
Hole mobility (cm^2^/Vs)	5300	480	120	75	14	2	20	11	40	30	2
Dielectric constant	5.7	11.9	9.7	9.7	10.1	9.8	10	10.4	9.1	8.6	9.8
Breakdown field (10^5^ V/cm)	130	3	31	24	154		103	49.5			

**Table 4 materials-17-04522-t004:** The selected electronic/optical parameters of the perovskite material parameters.

Parameter	Level
Bandgap energy [eV]	1.5–2.5
PL quantum efficiency	70%
Absorption coefficient [cm^−1^]	10^4^–10^5^
Crystallization energy barrier [kJ/mol]	56.6–97.3
Charge carrier lifetime [ns]	>300
Carrier mobility [cm^2^/Vs]	≈50–200
Relative permittivity	3
Exciton	Wannier-type exciton
Exciton binding energy [meV]	<10
Trap-state density [cm^−3^]	10^10^ (single crystals) 10^15^–10^17^ (polycrystals)

**Table 5 materials-17-04522-t005:** The selected 300 K semiconductor material parameters for the photodetector’s fabrication.

Parameter	Si	Ge	GaN	GaAs	AlAs	InP	InGaAs	InAs	GaSb	AlSb	InSb	HgTe	CdTe
Group	IV	IV	III–V	III–V	III–V	III–V	III–V	III–V	III–V	III–V	III–V	II–VI	II–VI
Lattice constant (Å)/structure	5.431 (D)	5.658 (D)	3.189(a)/5.186(c) Wurtzite	5.653 (ZB)	5.661 (ZB)	5.870 (ZB)	5.870 (ZB)	6.058 (ZB)	6.096 (ZB)	6.136 (ZB)	6.479 (ZB)	6.453 (ZB)	6.476 (ZB)
Bulk moduls (Gpa)	98	75	172	75	74	71	69	58	56	55	47	43	42
Bandgap (eV)	1.124 (id)	0.660 (id)	3.39	1.426 (d)	2.153 (id)	1.350 (d)	0.735 (d)	0.354 (d)	0.730 (d)	1.615 (id)	0.175 (d)	−0.141 (d)	1.475 (d)
Electron effective mass	0.26	0.39	0.20	0.067	0.29	0.077	0.041	0.024	0.042	0.14	0.014	0.028	0.090
Hole effective mass	0.19	0.12	0.80(H)	0.082(L) 0.45(H)	0.11(L) 0.40(H)	0.12(L) 0.55(H)	0.05(L) 0.60(H)	0.025(L) 0.37(H)	0.4	0.98	0.018(L) 0.4(H)	0.40	0.66
Electron mobility (cm^2^/Vs)	1450	3900	1400	8500	294	5400	13,800	3 × 10^4^	5000	200	8 × 10^4^	26,500	1050
Hole mobility (cm^2^/Vs)	505	1900	300	400	105	180	1000	500	880	420	800	320	104
Electron saturation velocity (10^7^ cm/s)	1.0	0.70	2.7	1.0	0.85	1.0	0.6	4.0	0.5		4.0		
Thermal cond. (W/cmK)	1.31	0.31	1.3	0.5	0.8	0.7		0.27	0.4	0.7	0.15		0.06
Relative dielectric constant	11.9	16.0	8.9	12.8	10.0	12.5		15.1	15.7	12.0	17.9	21	10.2

ZB—zincblende, D—diamond, d—direct, id—indirect, H—heavy hole, L—light hole.

**Table 6 materials-17-04522-t006:** $\sqrt{\alpha \tau }$ product for selected IR materials at 300 K (after Ref. [74]).

IR Material	Parameter					$\sqrt{\alpha \tau }$ [(s/cm)^1/2^]	
	Doping Concentration	Absorption Coefficient		Carrier Lifetime		MWIR	LWIR
		MWIR	LWIR	MWIR	LWIR		
HgCdTe	5 × 10^13^ cm^−3^	3.2 × 10^3^ cm^−1^	2.2 × 10^3^ cm^−1^	10 ms	0.5 ms	5.66	1.05
T2SLs InAs/GaSb	5 × 10^14^ cm^−3^	2.4 × 10^3^ cm^−1^	1.6 × 10^3^ cm^−1^	20 ns	10 ns	6.9 × 10^−3^	4.0 × 10^−3^
T2SLs InAs/InAsSb	5 × 10^14^ cm^−3^	1.2 × 10^3^ cm^−1^	8.0 × 10^2^ cm^−1^	25 μs	5 μs	1.7 × 10^−1^	6.3 × 10^−2^

**Table 7 materials-17-04522-t007:** Parameters of the common TIs.

Parameters	Bi_2_Se_3_	Sb_2_Te_3_	Bi_2_Te_3_
Electron mobility (cm^2^/Vs)	600	275	1140
Hole mobility (cm^2^/Vs)	-	360	680
Bandgap (eV)	0.35	0.30	0.21
Refraction index	12.2–2.5 (500–900 nm)	1.2–4.2 (400–800 nm)	1.4–3.5 (400–800 nm)
Carrier type (undoped)	n-type	p-type	p-type
Thermal conductivity (W/mK)	2.40	1.65	3.00
Melting point (°C)	706	662	585
Absorption coefficient (cm^−1^)	10^3^	10^4^–10^6^	-

**Table 8 materials-17-04522-t008:** The performance of the CQD photovoltaic sensors.

Parameter	PbS CQD				HgTe CQD
	Emberion (Espoo, Finland) https://www.emberion.com/ (accessed on 25 August 2024) Ref. [132]	IMEC (Leuven, Belgium) https://www.imec-int.com/en (accessed on 25 August 2024) Ref. [124]	SWIR V.S. (Somerville, MA, USA) https://www.swirvisionsystems.com (accessed on 25 August 2024) Ref. [131]	STM (Husseren-Wesserling, Francja) https://www.st.com (accessed on 25 August 2024) Ref. [133]	Beijing Inst. Tech. (Beijing China) https://www.bit.edu.cn (accessed on 25 August 2024) Ref. [112]
	** 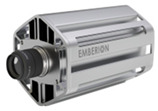 **	** 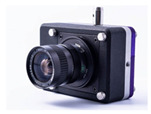 **	** 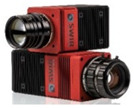 **	** 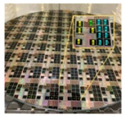 **	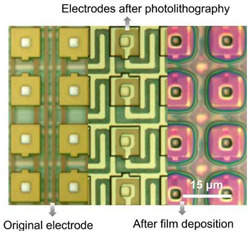
Resolution	640 × 512	768 × 512	1920 × 1080	0.9 MPx	1280 × 1024
Pixel pitch (μm)	20	5	15	1.62	15
*λ_peak_* (nm)	1850	1450	1470	1400	2
EQE (%)	20	40	15	60	14
Dark current (μA/cm^2^)		3.3@RT	NA	0.25@60 °C	≈6@RT
Dynamic range	80	82	70	54.4	NA

## Data Availability

Dataset available on request from the authors.

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
