# Peer review of "Performance of Low-Dimensional Solid Room-Temperature Photodetectors—Critical View"

_materials, 2024, doi:10.3390/ma17184522_

Round 1

Reviewer 1 Report

Comments and Suggestions for Authors

The authors compile a detailed review of the next generation of photodetectors using low-dimensional materials.

The review is very well organized, easy to follow, and will be of interest to many readers.

There were no outstanding questions.

I recommend accepting this excellent review paper as it stands.

I have one small question.

1. In Figure 17, is the blue line for Gr/ZnO QDs the result of fitting, I understand that the 5-point plot is a straight line, but you would need to specify why it is curved at the upper left over 106.

Reviewer 2 Report

Comments and Suggestions for Authors

This is a very interesting manuscript presenting a comprehensive review of the advancements in low-dimensional solid room-temperature photodetectors, emphasizing the significance of new materials and their potential applications. In my opinion, this manuscript is ready for publication after minor revisions:

  1. The discussion on the significance of photon IR detectors operating at 300 K is compelling. However, the authors should elaborate on the specific applications that would benefit from this technology.

  2. The authors' mention of misreading results due to device characterization issues is crucial. A brief discussion of how these issues can be mitigated would add depth to this section.

  3. The section on CQD layers and their integration with ROIC is well-articulated. However, including examples of successful implementations could provide readers with practical insights.

Reviewer 3 Report

Comments and Suggestions for Authors

This review paper delves into the advancements and performance of low-dimensional solid photodetectors, evidencing their potential to outperform traditional photodetectors across various spectral ranges. The authors explore the unique properties of materials like 2D materials and perovskites, discussing their strengths and challenges, especially for near-room-temperature applications. 

Below are some points to be considered:

1. Use more descriptive and engaging headings to make the manuscript easier to navigate. Clear headings help readers quickly find the information they are looking for.

2. Provide a more detailed discussion on the challenges faced by LDS photodetectors, such as defect sensitivity and environmental instability.

3. Can the authors provide the methodology of the inclusion/exclusion criteria of the works considered in this review?

4. Enhance the discussion on the market potential and commercial viability of LDS photodetectors.

Comments on the Quality of English Language

Use a more engaging and accessible writing style to appeal to a broader audience. This includes non-specialists who might be interested in the technological implications of LDS photodetectors.

Reviewer 4 Report

Comments and Suggestions for Authors

Specific comments and recommendations:

1) In my opinion, the title of this work is not attractive enough for potential readers. This manuscript is a review and as such should attract the attention of readers. For example, what exactly is the performance of electronic components and why is room temperature specified in the title? The word "Performance" occurs too often in the manuscript.

2) Some figures, such as Figure 10C, Figure 13, and Figure 15 are difficult to read. Their quality needs to be improved. Also, the formulas are in smaller fonts and are hard to read. All formulas must be written according to the journal standard;

3) For all figures, the authors have to check whether they also need written consent from publishers for image use;

4) To the best of my knowledge, perovskite materials were not created before four years ago, and the authors should provide a brief background on the research and creation of these materials. It is good, in addition to advantages, to point out some disadvantages of this type of material. The authors of the manuscript should emphasize the development trends and application areas from an engineering perspective.

Comments on the Quality of English Language

Minor editing of English language required.
